# Finding Competence Regions in Domain Generalization

**Jens Müller**  *jens.mueller@iwr.uni-heidelberg.de*
*Informatics for Life, Heidelberg University, Germany*

**Stefan T. Radev**  *stefan.radev93@gmail.com*
*STRUCTURES Cluster of Excellence, Heidelberg University, Germany*

**Robert Schmier**  *robert.schmier@de.bosch.com*
*Bosch Center for Artificial Intelligence, Renningen, Germany*
*Heidelberg University, Germany*

**Felix Draxler**  *felix.draxler@iwr.uni-heidelberg.*
*Heidelberg University, Germany*

**Carsten Rother**  *carsten.rother@iwr.uni-heidelberg.de*
*Heidelberg University, Germany*

**Ullrich Köthe**  *ullrich.koethe@iwr.uni-heidelberg.de*
*Heidelberg University, Germany*

**Reviewed on OpenReview:** *https://openreview.net/forum?id=TSyOvuwQFN*

## Abstract

We investigate a "learning to reject" framework to address the problem of silent failures in Domain Generalization (DG), where the test distribution differs from the training distribution. Assuming a mild distribution shift, we wish to accept out-of-distribution (OOD) data from a new domain whenever a model's estimated competence foresees trustworthy responses, instead of rejecting OOD data outright. Trustworthiness is then predicted via a proxy *incompetence score* that is tightly linked to the performance of a classifier. We present a comprehensive experimental evaluation of existing proxy scores as incompetence scores for classification and highlight the resulting trade-offs between rejection rate and accuracy gain. For comparability with prior work, we focus on standard DG benchmarks and consider the effect of measuring incompetence via different learned representations in a closed versus an open world setting. Our results suggest that increasing incompetence scores are indeed predictive of reduced accuracy, leading to significant improvements of the average accuracy below a suitable incompetence threshold. However, the scores are not yet good enough to allow for a favorable accuracy/rejection trade-off in all tested domains. Surprisingly, our results also indicate that classifiers optimized for DG robustness do not outperform a naive Empirical Risk Minimization (ERM) baseline in the competence region, that is, where test samples elicit low incompetence scores.

## 1 Introduction

Although modern deep learning methods exhibits excellent generalization, they are prone to silent failures when the actual data distribution differs from the distribution during training (Sanner et al., 2021; Yang et al., 2021). We address this problem in a "learning to reject" framework (Flores, 1958; Hendrickx et al., 2021; Zhang et al., 2023): *Given a pre-trained model and potentially problematic data instances, can we determine if the model's responses are still trustworthy?*

A major goal of this work is to explore the above question in settings where we wish to make predictions on a test set from a *new domain* following a potentially different distribution than the one available during training. This setting is referred to as Domain Generalization (DG) and assumes that we have access to multiple domains (also known as data sets or environments) during training. The generalization task asks to provide accurate predictions for a new domain, usually subject to a mild distribution shift (e.g., from one hospital to the next). Still, from a data-centric perspective, almost all instances in the new domain are out-of-distribution (OOD). Following the rationale of DG, we do not want to reject all OOD instances outright, but only those for which the estimated model competence falls below some acceptance threshold.

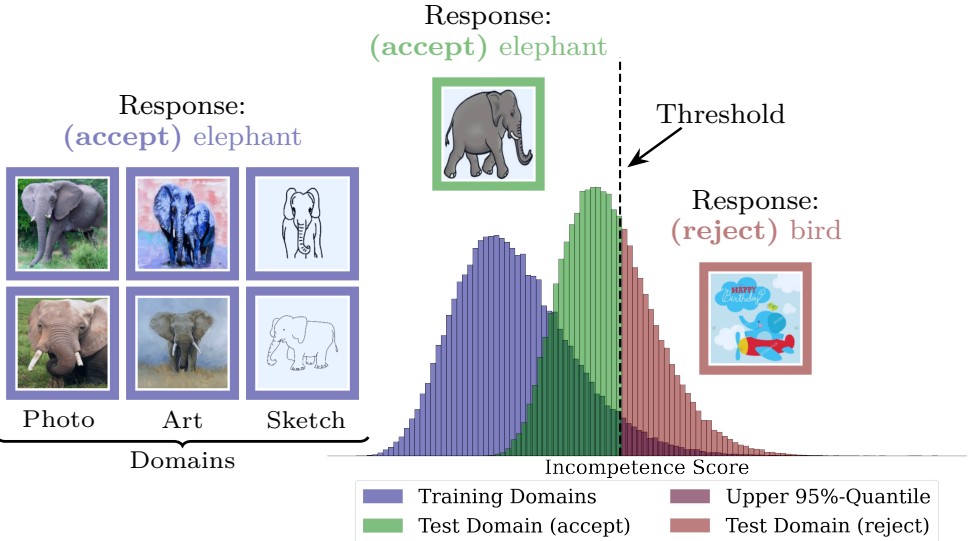

Figure 1: The main principle behind *incompetence scores* for improved domain generalization: We reject instances above the incompetence threshold, which is located at the 95% quantile of the training distribution.

Since we do not have access to the distribution of the test data during training, we can neither determine out-of-domain competence directly (Xie et al., 2006), nor define the acceptance threshold in a Bayes-optimal way (Chow, 1970). Instead, we investigate proxy scores that are negatively correlated with competence: We call them *incompetence scores* and they should monotonically decrease as a model's accuracy increases (see Section 3). For a simple example of such a score, we may consider the distance of a new data point to the nearest neighbor of the training data in a model's learned feature space. In this case, we expect the performance to drop with increasing distance. Interestingly, our experiments demonstrate that the monotonicity property typically holds for well-known choices of these scores.

Setting a threshold to delineate a *competence region* inevitably results in a trade-off between accuracy and coverage: The more instances we add to the competence region, the worse the accuracy, and *vice versa*. Identifying the optimal (task-dependent) trade-off in DG is difficult due to the differences between the training and the (unknown) test distributions. Thus, we find it pertinent to explore this trade-off for different thresholds across DG tasks (see Section 4.3).

The concept of incompetence underlying the present work is strongly linked to previous research on classification with a reject option (e.g., Bartlett & Wegkamp, 2008) and selective classification (e.g., Geifman & El-Yaniv, 2017). Common to these concepts is the idea to accurately predict errors based on a proxy quantity. Since we are interested in the task-dependent competence of pre-trained classifiers, we concentrate solely on post-hoc OOD detection methods as proxy scores for incompetence. And although the current work focuses on classification tasks, our approach should also be worth pursuing in regression tasks, since feature-based incompetence scores seem equally applicable in this case.

In the following, we present a comprehensive experimental evaluation of incompetence scores in a variety of DG tasks. For comparability with prior work, we focus on standard data sets from the DG literature

(Gulrajani & Lopez-Paz, 2020) and consider the closed vs. open world setting (i.e., new appearances of known classes vs. hitherto unknown classes) as well as the effect of measuring incompetence through different data representations. Further, we investigate whether state-of-the-art classifiers that are optimized specifically for domain shift robustness exhibit more accurate competence regions than naively trained ones. Finally, we investigate whether it is possible to estimate an incompetence threshold, such that a classifier is guaranteed to recover its ID accuracy in the corresponding competence region under domain shift. In summary, we make the following contributions:

1. We demonstrate empirically that accuracy decreases as incompetence scores increase and highlight the resulting trade-offs between rejection rate and accuracy gain (see Section 4.3)

2. We find that both feature- and logit-based scores are competitive in the closed world, whereas feature-based approaches work best in the open world setting (see Section 4.4 and Section 4.5)

3. We propose an approach to determine an incompetence threshold from ID data and demonstrate its utility for most domain shifts considered in this work (see Section 4.6)

4. We observe that robust classifiers do not outperform a naive baseline in terms of generalization performance in the elicited competence regions (see Section 4.4)

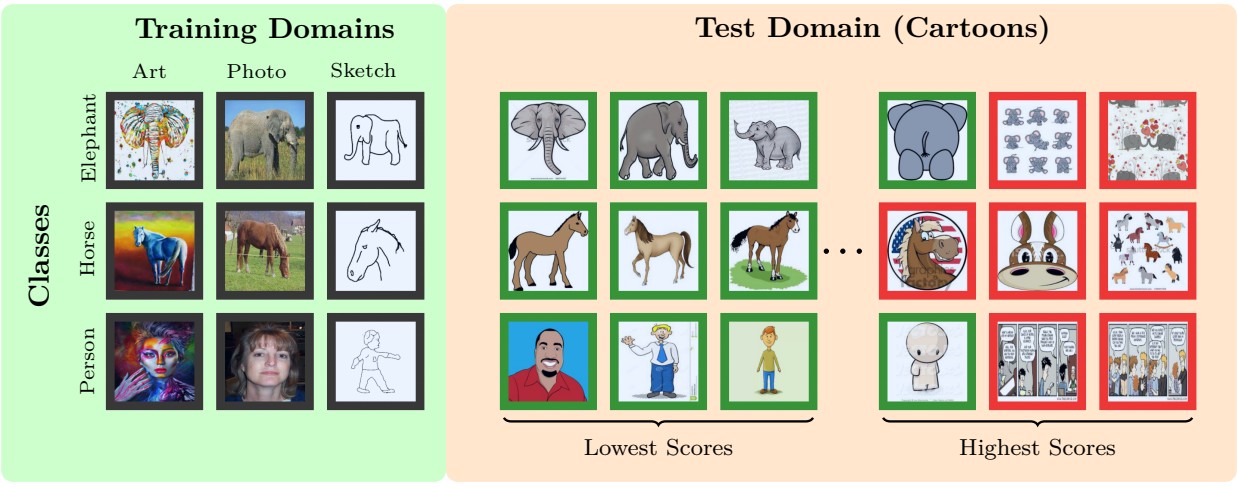

Figure 2: An incompetence score is able to sort out-of-distribution (OOD) images from the PACS data set, so that higher incompetence scores result in lower classification accuracy. *(Left)* Example images from the training domains. *(Right)* Images from the test domains resulting in lowest and highest incompetence scores (using a Deep-KNN scoring function) in the feature space of a baseline ERM classifier. Green and red frames denote correctly and incorrectly classified images, respectively. Higher incompetence scores correlate with a decrease in the classifier's accuracy.

## 2 Related Work

### 2.1 OOD Detection

Dealing with anomalous (i.e., out-of-distribution; OOD) instances that differ from those contained in the training set (i.e., our proxy for the in-distribution; ID) is a widely discussed and conceptually overloaded topic in the machine and statistical learning literature (Aggarwal & Yu, 2001; Yang et al., 2021; Shen et al., 2021; Han et al., 2022; Yang et al., 2022). OOD detection addresses the problem of flagging unusual data points which could undermine the reliability of machine learning systems (Yang et al., 2021); OOD generalization addresses the need to make predictions even when the test distribution is completely unknown or known to be different than the training distribution (Shen et al., 2021).

In this work, we are interested in analyzing established domain-robust classifiers. Thus, we focus on OOD detection methods that do not modify the classifier architecture or training. Such methods are called *post-hoc* detection (Yang et al., 2021), as they do not intervene on the downstream classifier. In this work, we utilize established post-hoc methods that rely on various aspects of model output such as the softmax output (e.g. Hendrycks & Gimpel (2016)), logit output (e.g. Hendrycks et al. (2019); Liu et al. (2020)), or intermediate feature-outputs (e.g. Aggarwal & Aggarwal (2017); Sun et al. (2022); Wang et al. (2022)).

Post-hoc OOD scores have been shown to perform well across a variety of OOD detection benchmarks (Yang et al., 2022). Previous work analyzed post-hoc OOD detection scores to predict the accuracy of a classifier on novel inputs (Techapanurak & Okatani, 2021) or to detect ID failure cases (Xia & Bouganis, 2022). In addition, Techapanurak & Okatani (2021) compute an aggregated OOD-score over an entire ID data set to predict the global accuracy of a classifier on OOD data. Differently, we aim to predict the likelihood of error from individual incompetence score values and show that this approach provides us with a finer control over the trade-off between coverage and accuracy (see Section 4.6).

Despite the large volume of literature focusing on OOD detection and generalization (e.g., Ruff et al. (2021)), there are no extensive studies applying OOD scores to domain generalization (DG) benchmarks. Thus, one of the main goals of this work was to provide such a comprehensive analysis on the utility of OOD scores for improving DG.

## 2.2 Domain Generalization

The goal of domain generalization (DG) is to train models that generalize well under covariate shifts (Muandet et al., 2013; Zhou et al., 2022), such as adversarial attacks (Goodfellow et al., 2014) or style changes (Gatys et al., 2016), for which the label space remains unchanged during testing (Yang et al., 2021). In DG settings, we assume that we have access to different environments or data sets (e.g., art and sketch images) and the goal is to make good predictions in completely unknown environments (e.g., real world images).

Compared to Domain Adaptation (DA; Wang & Deng, 2018), where we have unlabeled data from the test domain, the DG problems assume that we have no knowledge about the test domain(s). Consequently, it is not possible to train the algorithm using unlabeled test data as in self-training (Xiaojin, 2008). However, a recent study has demonstrated that a model can be effectively adjusted during test time (Chen et al., 2023). Moreover, it has been shown that classifiers can assign high likelihoods under domain shift even when they are plainly wrong, which makes it hard to detect failure cases (Nguyen et al., 2015; Nalisnick et al., 2019). Thus, proxy "incompetence" OOD scores appear to be good candidates for spotlighting such failures. However, to the best of our knowledge, there are no extensive studies which attempt to quantify the competence of domain-robust models in the context of DG.

Many benchmark data sets in DG have been established, on which researchers can study generalization performance beyond a single training environment (Gulrajani & Lopez-Paz, 2020; Koh et al., 2021). In this work, we consider the main data sets contained in the DomainBed benchmark (Gulrajani & Lopez-Paz, 2020). We additionally distinguish between a *closed world* setting, where only instances of known classes are encountered in the test domain, and an *open world* setting, where instances of unknown classes are also present in the test domain. We believe the open world setting to be of practical interest, even though typical DG problems are formulated under a closed world assumption (Zhou et al., 2022).

Domain generalization approaches can roughly be classified into two categories: learning-based methods and augmentation-based methods (Li et al., 2021). The current work primarily focuses on learning-based approaches that seek to learn features that remain invariant under domain shift. According to Ben-David et al. (2006), there is theoretical evidence to suggest that features that remain invariant across domains enable accurate predictions in cases of distributional shifts. As a result, various algorithms have been proposed with the goal of learning invariant features (Muandet et al., 2013; Arjovsky et al., 2019; Müller et al., 2021; Shi et al., 2021). However, it is not clear which DG methods can achieve consistently robust performance across different data sets. On the one hand, it has been suggested that a strong standard classifier trained with empirical risk estimation (ERM) performs favorably across multiple DG data sets (Gulrajani & Lopez-Paz, 2020; Korevaar et al., 2023). On the other hand, some DG methods have been shown to outperform an ERM baseline on several benchmark data sets (Koh et al., 2021). Here, we complement the existing literature

by examining whether the competence regions of different DG classifiers differ in terms of the achieved improvements in accuracy.

## 2.3 Selective Classification

Inference with a reject option (aka *selective classification*, Geifman & El-Yaniv, 2017; El-Yaniv et al., 2010) enables classifiers to refrain from making a prediction under ambiguous or novel conditions (Hendrickx et al., 2021). The reject option has been extensively studied in statistical and machine learning (Hellman, 1970; Fumera & Roli, 2002; Grandvalet et al., 2008; Wegkamp & Yuan, 2012). The origins of these approaches can be traced back until at least the 50s of the last century, as demonstrated by works such as Chow (1957; 1970); Hellman (1970). However, selective classification has only recently gained attention in the context of deep neural networks (Geifman & El-Yaniv, 2017).

Zhang et al. (2023) outline the three main reasons why a reject option could be a reasonable choice in any practical application: 1) failure cases; 2) unknown cases; and 3) fake inputs. For instance, Kamath et al. (2020) train natural language processing (NLP) models for selective question answering under domain shift. Varshney et al. (2022) investigate the utility of MaxProb (a common OOD detection score) as a rejection criterion across several NLP data sets. Ren et al. (2022) use the Mahalanobis distance as OOD detection method to filter inputs to NLP models for conditional text generation and Mesquita et al. (2016) showcase the reject option for catching software defects.

The main challenge selective classifiers face is how to reduce the error rate by "rejecting" instances for which no reliable prediction can be made, while keeping coverage (i.e., the number of "accepted" instances) as high as possible (Condessa et al., 2017; Nadeem et al., 2009; Chow, 1970). And while the theoretical characteristics of the resulting trade-off have been systematically studied (El-Yaniv et al., 2010; Wiener & El-Yaniv, 2011), the empirical utility of OOD "rejection scores" for ensuring robust performance in the DG setting remains largely unclear. In this work, we perform an extensive evaluation of this trade-off across a wide variety of state-of-the-art OOD scores, domain-robust classifiers, DG data sets and environments.

## 3 Method

We denote with $c_\theta$ an arbitrary classifier with a vector of trainable parameters $\theta$ (e.g., neural network weights) which we typically suppress for readability. To evaluate a classifier, we consider its accuracy, which we denote as $A_{dist}$, based on inference queries from some reference distribution $x \sim p_{dist}(x)$.

### 3.1 Incompetence Scores

The goal of an incompetence score $s_c \colon \mathbb{R}^D \to \mathbb{R}$ is to indicate whether a classifier $c$ is familiar with some input $x \in \mathcal{X}$. We consider familiarity with the input to be equivalent to competence. The fundamental principle of this work is that instances eliciting a high incompetence score are intrinsically hard to predict and *vice versa*. Due to the close conceptual connection between competence and familiarity or incompetence and OOD, we employ OOD scores as proxy for incompetence. In particular, we employ *post-hoc* methods that compute an OOD score taking into account the classifier.

In our subsequent experiments, we compute the incompetence scores via a number of *post-hoc* methods. The *post-hoc* methods used in this paper can be grouped into the following categories:

- Feature-based: Virtual-logit Matching (ViM; Wang et al., 2022), Deep-KNN (Deep-KNN; Sun et al., 2022);

- Density-based: Gaussian mixture models (GMM), minimum Mahalanobis distance between features and class-wise centroids (Mahalanobis; Lee et al., 2018);

- Reconstruction-based: reconstruction error of PCA in feature space (Aggarwal & Aggarwal, 2017);

- Logit-based: energy score (Energy; Liu et al., 2020), maximum logit (Logit; Hendrycks et al., 2019), maximum softmax (Softmax; Hendrycks & Gimpel, 2016), and energy-react (Energy-React; Sun et al., 2021).

Note, that we interpret higher scores as indicative of incompetence (e.g., we consider the negative of the maximum softmax and the maximum logit).

## 3.2 Admissible Incompetence Scores

Detecting out-of-competence means checking whether some given incompetence score $s_c(x) \in \mathbb{R}$ falls below some threshold $\alpha$ (classified as in-competence) or above (classified as out-of-competence). We consider scores $s_c(x)$ that depend on the classifier $c$ and the input $x$ at hand.

The threshold $\alpha$ trades off accuracy (how well does the classifier perform on accepted data) with coverage (how many samples does the score accept). In this section, we describe how a useful (ideal) incompetence score should affect downstream classification as a function of the threshold $\alpha$. In particular, consider the subset of input space where the classifier is deemed competent given a fixed threshold $\alpha$:

$$X_c(\alpha) := \{x : s_c(x) \leq \alpha\}. \tag{1}$$

We use the ID data to determine a suitable threshold for the competence region, for instance we later pick $\alpha = \alpha_{95}$ such that 95% of the ID data is in $X_c(\alpha_{95})$. We consider the accuracy $\mathrm{A_{OOD}}(\alpha)$ of the classifier $c$ on the unknown test domain restricted to the competence region $X_c(\alpha)$ as a function of $\alpha$.

We summarize the above description in the fundamental criterion of this work: **An admissible incompetence score must assign low incompetence to those regions where the downstream accuracy is high.** We formalize this as follows:

**Criterion 3.1.** *An incompetence score $s_c(x)$ is called "admissible" if the downstream accuracy $\mathrm{A_{OOD}}(\alpha)$ decreases monotonically as $\alpha$ is increased for any distribution that undergoes a mild covariate shift.*

This monotonic trend requires that the incompetence score $s_c(x)$ is closely related to the performance of the classifier. Such a connection allows us to make predictions on the downstream accuracy as a function of $\alpha$:

**Proposition 3.1.** *Given a classifier $c_\theta(x)$ and its corresponding in-distribution $p_{\mathrm{ID}}$. Then, for a test distribution of interest $p_{\mathrm{OOD}}$ and a corresponding admissible score $s_c(x)$ as in Criterion 3.1:*

  (a) *If there is a threshold $\alpha^* \in \mathbb{R}$ such that for all $\alpha \leq \alpha^*$ ID and OOD have the same support and classification accuracy, then, $\mathrm{A_{OOD}}(\alpha) \geq \mathrm{A_{ID}}$ for $\alpha < \alpha^*$.*

  (b) *In the limit of $\alpha \to \infty$, we find that $\mathrm{A_{OOD}}(\alpha) \to \mathrm{A_{OOD}}$.*

The first statement describes the behavior of $\mathrm{A_{OOD}}(\alpha)$ for small $\alpha$ and the second for large $\alpha$. We observe this behavior empirically in Figure 3 and give the proof in Appendix A.7

## 4 Experiments

In our experiments, we analyze the effect of an incompetence threshold $\alpha$ (see Section 3.2) on DG performance.[1] In the following, we first describe our experimental protocol. Then, we analyze the competence region in dependence on the threshold $\alpha$ and show that the competence region behaves as predicted in Proposition 3.1. Finally, we carry out an extensive investigation of the competence region for closed and open world settings, where we show the utility of the concept for various incompetence scores and point out current weaknesses.

As an introductory example to the competence region, we consider Figure 2 which depicts the experimental procedure on the PACS data set for a standard classifier trained with Empirical Risk Minimization (ERM;

---

[1]We provide access to our code under https://github.com/XarwinM/competence_estimation

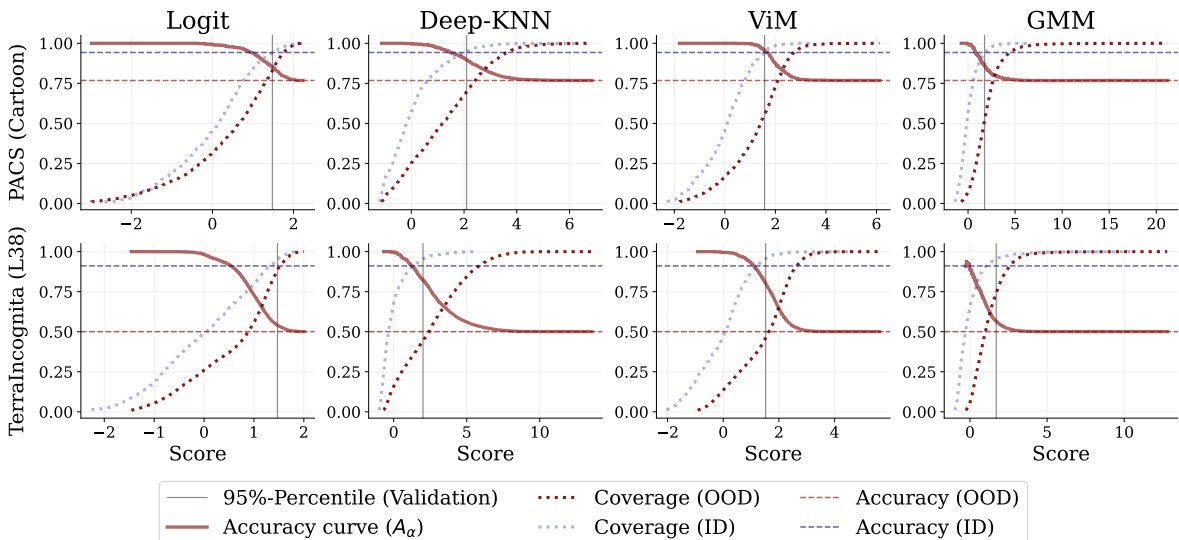

Figure 3: The accuracy of the ERM classifier on OOD data $A_{OOD}(\alpha)$ as the competence region is enlarged by increasing the incompetence threshold $\alpha$. As predicted by our monotonicity criterion (Criterion 3.1), the accuracy starts off at $A_{OOD}(\alpha) \geq A_{ID}$ and then falls off monotonically with $\alpha$. At the same time, the fraction of data the classifier is applied to increases. The classifier accuracy and fraction of considered data can easily be traded off using this figure.

Vapnik, 1999). We train the classifier on the domains Art, Photo, and Sketch, and apply the trained classifier in the unknown Cartoon domain. The samples in the test domain are ordered by the predicted incompetence score $s_c(x)$. As expected, the classifier still performs well on Cartoon samples with low incompetence scores (9 out of 9 classified correctly in the example), but the accuracy drops for high scores (only 2 out of 9 correct classification). Qualitatively, the score correctly notices that images with significantly different characteristics are much harder to classify. In the following sections, we quantify this behavior systematically for a number of different classifiers, incompetence scores, data sets, and domain. But first, we give details on our experimental setup.

## 4.1   Experimental Setup

We consider all combinations of nine pre-trained classifiers $c_\theta(x)$, varying both in architecture and training, nine OOD post-hoc scores $s_c(x)$ as incompetence scores on a total of 32 DG tasks from six different DG data sets. The pre-trained classifiers are obtained as follows. We train various state-of-the-art classifiers from DG literature, namely Fish (Shi et al., 2021), GroupDRO (Sagawa et al., 2019), SD (Pezeshki et al., 2021), SagNet (Nam et al., 2021), Mixup (Yan et al., 2020) and VREx (Krueger et al., 2021). Furthermore, we train three different neural network architectures with empirical-risk-minimization (Vapnik, 1999): A ResNet based architecture which we denote by ERM (He et al., 2016), a Vision Transformer (Dosovitskiy et al., 2020) and a Swin Transformer (Liu et al., 2021). Training details and hyperparameter settings are listed in Appendix A.5.

These models are trained on six domain generalization data sets from the DomainBed repository (Gulrajani & Lopez-Paz, 2020): PACS (Li et al., 2017), OfficeHome (Venkateswara et al., 2017), VLCS (Fang et al., 2013), TerraIncognita (Beery et al., 2018), DomainNet (Peng et al., 2019) and SVIRO (Cruz et al., 2020). Each DG data set consists of four to ten different domains from which we construct different DG tasks: We train a classifier on all but one domain. The one left out during training is then the OOD test domain where the competence region is evaluated. As an example consider the DG task behind the earlier example in Figure 2: If we train a model on the domains Photos, Art images, and Sketches, the DG task asks for an accurate model on the domain Cartoons which constitute the OOD test domain (see Figure 2). Overall we

consider 32 DG tasks which result in 288 trained networks. We then compute the incompetence scores of each trained network In Section 3.1, we describe the process of calculating the incompetence scores.

For each DG task, we distinguish four data sets. For the ID distribution, we consider a training set, a validation set for hyperparameter optimization, and a test set that has no influence on the optimization process for the subsequent evaluation. The classifiers are trained on the ID training set. We compute the scores for the ID distribution on the ID validation set and the ID accuracy on the ID test set. The OOD test set is given by the DG task (e.g., as in Figure 2). After training, we apply all post-hoc methods to the penultimate feature layer or the ouput (logits) layer of the classifier, as is typical in the OOD detection literature. If the post-hoc method needs to fit the data (as for instance with GMMs), we fit the score function on the ID training data.

## 4.2 Competence Threshold

In this section, we analyze the performance of the classifiers as a function of the threshold $\alpha$ which determines their competence region (see Equation 1 in Section 3). To this end, we compute the incompetence scores on the ID validation data set and on all OOD data samples.

Figure 3 depicts the resulting score distributions and accuracy $A_{OOD}(\alpha)$ as a function of the threshold $\alpha$ for a single classifier (ERM). Here, we consider four incompetence scores on one of the DG tasks provided by PACS and TerraIncognita, respectively. We find that the considered incompetence scores fulfill the requirement for a competence detector in Criterion 3.1 that the accuracy must decrease monotonically as the threshold $\alpha$ increases. We then find the theoretical results in Section 3 confirmed: For low $\alpha$, the accuracy $A_{OOD}(\alpha)$ is high, and even exceeds the average accuracy on the ID data $A_{ID}$ (see Proposition 3.1 (a)). It eventually decreases until $A_{OOD}(\alpha) \rightarrow A_{OOD}$ for large $\alpha$ (see Proposition 3.1b).

Figure 3 also depicts the fraction of ID and OOD data that is considered (i.e., not rejected) as we increase the incompetence threshold $\alpha$:

$$\text{Coverage}_{\text{dist}}(\alpha) = \frac{|\mathcal{D}_{\text{dist}} \cap X_c(\alpha)|}{|\mathcal{D}_{\text{dist}}|}. \tag{2}$$

For instance, we can compare the methods at the $\alpha_{95}$ which includes 95% of the ID data (vertical gray line in Figure 3). Here, Logit keeps a significantly larger fraction of test data compared to the other incompetence scores. However, this results in a lower accuracy in the competence region $A_{OOD}(\alpha_{95})$.

Unfortunately, due to the nature of the DG problem, the accuracy curve $A_{OOD}(\alpha)$ in Figure 3 is not accessible during inference, which makes it difficult to choose a suitable threshold $\alpha$. In Figure 3 we can observe that ViM and KNN achieve at the 95% percentile (w.r.t. the ID validation set) an accuracy that is comparable to the ID accuracy, rendering the predictions in this competence region very accurate and trustworthy. GMM and Logit obtain very high accuracies in the competence region $X_c(\alpha) \cap \mathcal{D}_{OOD}$ for small $\alpha$ values, but exhibit a larger drop in accuracy at the 95% percentile (w.r.t. the ID distribution). We show the accuracies $A_{OOD}(\alpha)$ for different threshold values $\alpha$ for all data sets and DG tasks in Appendix A.2.

## 4.3 Accuracy vs. Coverage Trade-Off

We illustrate the trade-off between accuracy and coverage for the ViM, GMM and Logit score for all domains in PACS and TerraIncognita in Figure 4. Here, we consider the empirical average over all classifiers. All scores behave relatively monotonically in the sense that increased coverage results in a reduction in accuracy.

First, it is evident that GMM (red curve) shows a non-competitive accuracy-coverage trade-off. Further, while the Logit score (green curve) exhibits a slightly favorable accuracy coverage trade-off across the PACS domain, a clear winner for TerraIncognita does not emerge. Overerall, ViM (blue curve) performs better than the Logit score in terms of accuracy in the competence region elicited via a threshold at the 95[th] percentile of the score ID distribution. This indicates that the ViM score demonstrates greater selectivity and classifies more samples as outliers in comparison to the Logit score. However, for all cases considered, we conclude that Logit and ViM exhibit a similar accuracy coverage trade-off. Note that the curves in Figure 4 are not accessible when we need to set the threshold.

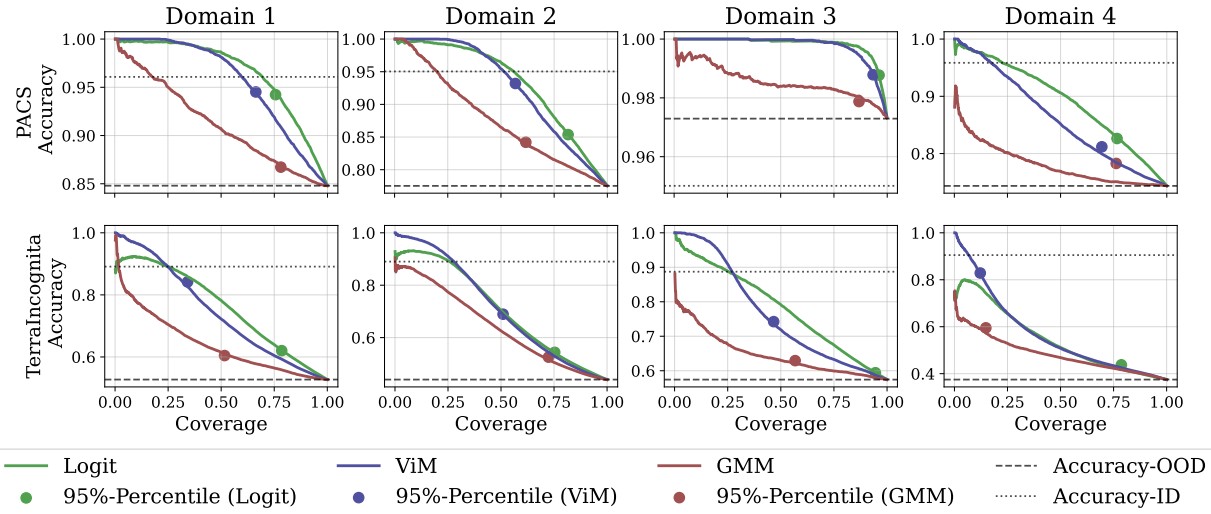

Figure 4: The expectation over the different domain robust classifier's accuracy on OOD data as a function of the coverage of the competence region. The 95$^{\text{th}}$ percentiles refer to the validation set of the ID distribution and also represent averages over all classifiers (and are thus sometimes off-curve). The domains are ordered in canonical sequence as in Table 2 in Appendix A.3.

## 4.4 Extensive Survey

| In Percentages (%) | PACS | | | OfficeHome | | | VLCS | | |
|---|---|---|---|---|---|---|---|---|---|
| | OOD-Gain ↑ | ID-Gain ↑ | Coverage ↑ | OOD-Gain ↑ | ID-Gain ↑ | Coverage ↑ | OOD-Gain ↑ | ID-Gain ↑ | Coverage ↑ |
| Deep-KNN | **11** [1-18] | **0** [-12-5] | 66 [56-95] | 8 [3-16] | -13 [-28-1] | 82 [64-94] | **2** [0-5] | -9 [-27-14] | 87 [72-99] |
| ViM | 9 [1-19] | **0** [-17-5] | 66 [50-93] | 5 [2-13] | -14 [-32-1] | 87 [65-95] | **2** [0-5] | **-8** [-28-14] | 85 [62-99] |
| Softmax | 7 [1-14] | -4 [-11-5] | 84 [65-97] | 8 [3-15] | **-12** [-32-1] | 84 [67-95] | **2** [0-4] | -10 [-27-13] | 93 [87-99] |
| Logit | 9 [1-12] | -3 [-12-5] | 80 [61-96] | **9** [2-16] | -13 [-33-0] | 81 [66-96] | **2** [0-5] | -10 [-27-14] | 92 [83-98] |
| Energy | 9 [1-12] | -3 [-12-5] | 79 [61-96] | 8 [2-16] | -14 [-33-0] | 82 [67-96] | **2** [0-4] | -10 [-27-14] | 93 [82-98] |
| Energy-React | 9 [1-12] | -3 [-13-5] | 79 [60-96] | 8 [2-16] | -14 [-33-0] | 82 [67-96] | **2** [0-4] | -10 [-27-13] | 93 [82-98] |
| Mahalonobis | 1 [0-12] | -8 [-22-4] | 80 [50-96] | 1 [0-7] | -17 [-42-0] | 91 [75-95] | 0 [-1-3] | -11 [-28-14] | 93 [73-99] |
| GMM | 2 [0-13] | -8 [-21-4] | 76 [50-96] | 0 [0-7] | -18 [-42-0] | 92 [76-95] | 0 [-1-3] | -12 [-28-14] | 85 [53-99] |
| PCA | 1 [-1-10] | -12 [-21-3] | 78 [57-97] | 0 [0-7] | -18 [-42-0] | 93 [78-96] | 0 [-1-2] | -12 [-28-14] | 88 [64-99] |

| | Terra Incognita | | | DomainNet | | | SVIRO | | |
|---|---|---|---|---|---|---|---|---|---|
| | OOD-Gain ↑ | ID-Gain ↑ | Coverage ↑ | OOD-Gain ↑ | ID-Gain ↑ | Coverage ↑ | OOD-Gain ↑ | ID-Gain ↑ | Coverage ↑ |
| Deep-KNN | **32** [12-51] | **-8** [-36-4] | 37 [13-52] | **4** [0-6] | **-6** [-50-7] | 85 [70-93] | **4** [1-28] | **0** [-1-0] | 28 [5-64] |
| ViM | 28 [13-51] | -13 [-35-3] | 41 [7-57] | 2 [0-7] | -8 [-50-6] | 90 [68-97] | **4** [1-30] | **0** [0-0] | 19 [6-62] |
| Softmax | 4 [1-12] | -38 [-54-24] | 85 [68-96] | 3 [0-5] | -8 [-52-5] | 94 [77-98] | **4** [0-24] | **0** [-10-0] | 60 [28-83] |
| Logit | 5 [1-18] | -34 [-55-23] | 85 [60-98] | 2 [0-5] | -8 [-51-6] | 93 [77-97] | 2 [0-21] | **0** [-19-0] | 67 [40-87] |
| Energy | 5 [1-19] | -33 [-55-21] | 85 [55-98] | 2 [0-5] | -9 [-51-5] | 94 [79-98] | 2 [-2-21] | **0** [-24-0] | 67 [40-87] |
| Energy-React | 5 [1-19] | -33 [-55-21] | 85 [55-98] | 2 [0-5] | -9 [-51-5] | 93 [79-98] | 2 [-2-21] | **0** [-24-0] | 67 [42-88] |
| Mahalonobis | 5 [-1-38] | -33 [-56-11] | 62 [7-94] | -1 [-3-5] | -11 [-53-4] | 92 [77-97] | 2 [-1-28] | **0** [-19-0] | 20 [5-95] |
| GMM | 7 [-1-38] | -28 [-51-10] | 56 [7-84] | -1 [-3-4] | -12 [-53-3] | 93 [79-98] | 3 [-1-28] | **0** [-11-0] | 20 [5-67] |
| PCA | 1 [-1-26] | -38 [-53-24] | 87 [35-99] | -1 [-3-2] | -12 [-53-2] | 92 [84-98] | 0 [-1-15] | -2 [-26-0] | 87 [47-99] |

Table 1: Accuracy on competence region of OOD domain for different domain generalization data sets and incompetence scores. As the threshold for the competence regions, we choose the 95% percentile of the ID validation set. For all metrics, a higher value means better performance (↑). All displayed values are medians over different domain roles and classifiers, brackets indicate 90% confidence interval.

In the following, we evaluate all nine incompetence scores on all six DG data sets using the nine classifiers. Since each data set features 32 different DG tasks, we perform a total of $32 \cdot 9 \cdot 9 = 2592$ experiments. For each experiment, we obtain accuracy curves as in Figure 2 as a function of $\alpha$. To summarize and compare the performance of each score on each data set, we need to deal with the trade-off between accuracy and coverage. Thus, we measure accuracy $\text{A}_{\text{OOD}}(\alpha_{95})$ at the score $\alpha_{95}$, such that 95% of ID validation data fall below this threshold, that is $\text{Frac}_{\text{ID}}(\alpha_{95}) = 95\%$. As mentioned in Section 4.2, choosing $\alpha$ in DG is

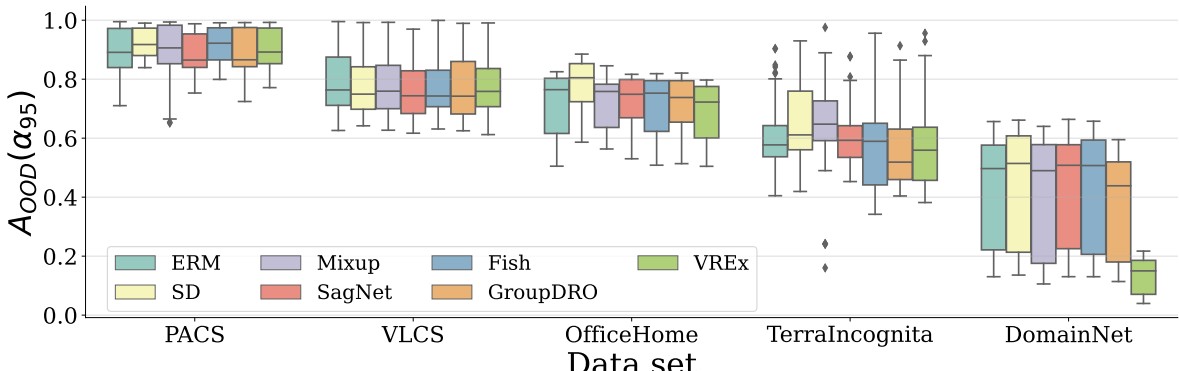

Figure 5: Accuracy in the competence region for different DG algorithms and OOD scores on different data sets. The chosen threshold corresponds to the $95^{\text{th}}$ percentile of the ID-distribution scores. Note: VREx failed to converge with the hyperparameters used on the DomainNet data set.

notoriously difficult, since we have no access to the test domain(s) during training. The following quantities provide useful summary statistics for comparing our results across all experiments:

1. OOD-Gain $= \text{A}_{\text{OOD}}(\alpha_{95}) - \text{A}_{\text{OOD}}$: The performance gain in the OOD domain by considering only the data in the competence region $X_c(\alpha_{95})$.

2. ID-Gain$= \text{A}_{\text{OOD}}(\alpha_{95}) - \text{A}_{\text{ID}}$: Expresses the performance gap between the accuracy on OOD data in the competence region $X_c(\alpha_{95})$ and the accuracy on the entire ID data $\text{A}_{\text{ID}}$.

3. Coverage $= \text{Coverage}_{\text{OOD}}(\alpha_{95})$ as given by Equation 2: The proportion of OOD data that falls within the competence region.

For each quantity, a higher value indicates better performance (↑). Note that the coverage of the competence region alone is not informative. A naive approach that includes all samples in the competence region would achieve the largest competence region but would fall short in terms of OOD-Gain or ID-Gain.

Table 1 summarizes the results from our extensive sweep over classifiers, data sets, and incompetence scores. The displayed values are the medians over different domain roles and classifiers. Overall, we observe that in the competence region, higher accuracy is achieved compared to the naive application on all OOD data instances. This confirms that incompetence score and accuracy are indeed tightly linked. However, for most DG data sets and incompetence scores, we are not able to replicate the ID accuracy. This indicates that we cannot naively expect the classifier to attain the same accuracy as observed in the ID distribution in the 95% percentile $\alpha_{95}$. Further important findings are:

- In general, feature-based (Deep-KNN, ViM), as well as logit-based incompetence scores (Softmax, Logit, Energy, Energy-React) obtain significantly higher accuracy on OOD data (higher OOD-Gain) by filtering the data to the competence region $X_c(\alpha_{95})$ than the density- and reconstruction-based approaches (Mahalanobis, GMM, PCA).

- The feature-based scores achieve a significant performance boost on TerraIncognita. TerraIncognita contains DG tasks that suffer from a particularly huge drop in accuracy from ID to OOD distribution (see Appendix A.6).

- The proportion of OOD data that falls inside the competence region (i.e., coverage) is smallest for feature-based methods, but they also provide the highest accuracy across all DG data sets.

It is important to note that at the specific threshold investigated, the accuracy in the competence region remains unaffected by the DG algorithms (see Figure 5). Based on this observation, the incremental utility of

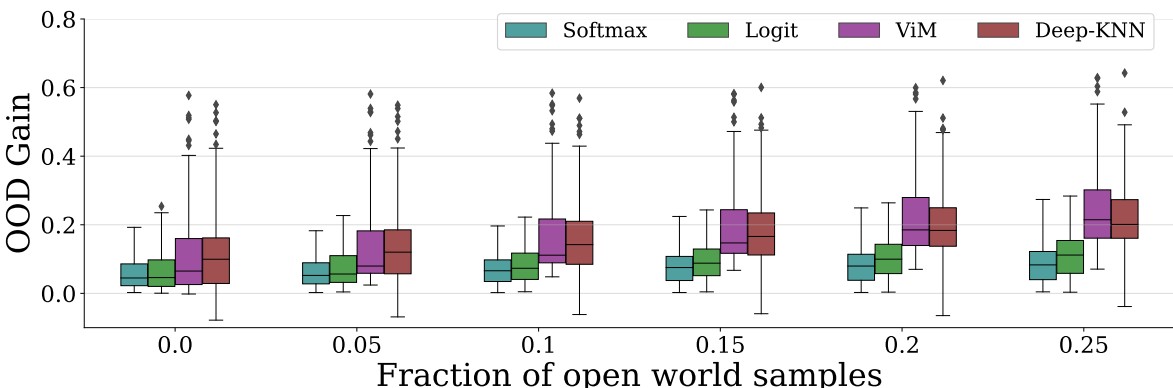

Figure 6: Performance of Logit and Softmax scores (logit-based) against Deep-KNN and ViM (feature-based) for an increasing fraction of open world data (unknown classes) in the test domain. The performance gain on the OOD data (*OOD-Gain*, higher is better) for the logit-based methods is less pronounced compared to ViM and Deep-KNN.

DG algorithms specifically designed for the purpose of domain robustness becomes uncertain. If we assume that DG algorithms successfully achieve their primary goal of learning domain-invariant features, then we can speculate that they create a more valuable competence region. However, this is not in line with our observation in Figure 5: We observe that no domain-robust classifier is consistently and significantly more accurate in the competence region than the simple baseline classifier (ERM). In Appendix A.6, we also show that the same result holds without restrictions on the competence region. Furthermore, we explore in Appendix A.2 thresholds close to the 95th percentile and demonstrated that the relative performance of the scores remains quite consistent.

### 4.5 Open World Performance

In this section, we study how different incompetence scores shape the competence region when instances of unknown classes are present in the ID distribution. Accordingly, for each domain in PACS, VLCS, Office-Home, and TerraIncognita data sets, we create a matching "open world" domain containing only instances of unknown classes. In total, we create 16 open world domains. For example, if we evaluate a model on the PACS Sketch domain, we create an open world domain containing only sketches of classes that are not in the PACS data set. We describe the procedure for creating the open world domains in detail in Appendix A.4. In the following, we restrict our analysis to the 16 domains for which an open world twin exists.

We enrich the existing test domains with 0%, 5%, 10%, 15%, 20%, and 25% instances with unknown classes. A good incompetence score should mark instances of unknown classes with a high value and therefore render them outside of the competence region $X_c(\alpha_{95})$. In this case, the OOD-Gain would increase as more open world instances find their way into the test set. In Figure 6 we observe that this behavior is achieved particularly well for the ViM score. The Logit and Softmax scores are less successful in delineating unknown class instances from the competence region and therefore the OOD-Gain is less pronounced.

Indeed, to test whether this observation holds statistically across all classifiers, we fit a hierarchical linear regression (Stephen & Anthony, 2002) on OOD Gain with `Classifier`, `Percentage Open World`, and `Incompetence Score`, as well as their interactions as fixed factors, together with `Data Set` and `Test Domain` as random factors (to account for the fact that the same classifier is evaluated in multiple data sets and test domains). The statistical results confirm the general trends visible in Figure 6. First, we find significant main effects of `Percentage Open World` (i.e., overall OOD Gain increases with an increasing number of open world instances) and `Incompetence Score` (i.e., ViM and Deep-KNN achieve a higher overall OOD Gain). Importantly, the only significant interaction revealed by the hierarchical regression model suggests that ViM is able to achieve the largest OOD Gain as the fraction of open world samples increases. Note, that the same trend is present for Deep-KNN, but it fails to achieve statistical significance due to its high

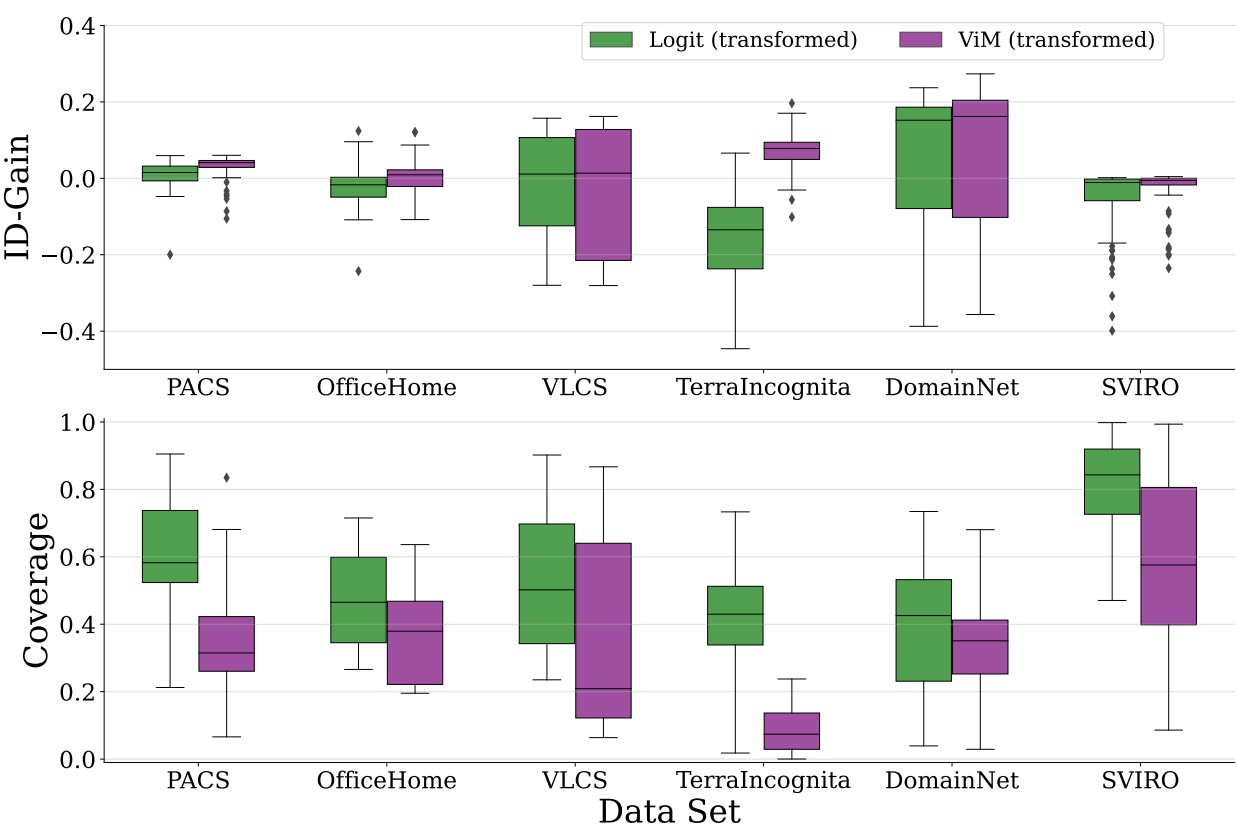

Figure 7: ID-Gain and Coverage for Logit and ViM if transformed as described in Section 4.6. The threshold is set such that the ID-Gain should be at least 0. *Top row:* ID-Gain for Logit and ViM due to different data sets. *Bottom row:* Coverage for Logit and ViM across different DG data sets. Medians and quantiles for all boxplots are computed over different domain roles and classifiers. The threshold is set as the ID accuracy.

variability (see Figure 6). Moreover, none of the effects involving the factor `Classifier` turn out to be significant predictors of OOD Gain, suggesting that the results are largely classifier-independent.

In the closed world setting, differences between logit- and feature-based scores are for most DG data sets small (see e.g. Table 1). However, we have shown that it is very relevant in the setting where instances of unknown classes occur. In Appendix A.4 we show the open world behavior for all incompetence scores.

### 4.6 Estimating the Incompetence Threshold

Choosing the 95% percentile of the ID distribution as incompetence threshold can be considered as weighting the trade-off between accuracy and coverage towards coverage – only 5% of ID data are rejected. We now seek a slightly different incompetence threshold which puts more weight on the accuracy. The question we want to address is whether *we can set a threshold such that a certain accuracy is achieved in the competence region?* This question is of high practical relevance, but also particularly challenging for two reasons. First, many scores used so far have no out-of-the-box connection to the accuracy and second, we deal with a domain shift that might result in a completely new score-accuracy relationship.

Thus, as a potential remedy, we suggest learning $\widetilde{s}_c(x) = p_{\text{ID}}(c(x) \neq y \mid s_c(x))$ and using this conditional probability as a *transformed* score. This score represents the probability of an incorrect prediction given the original score. If we define a competence region with an incompetence threshold of $1 - A_{\text{ID}}$, we can expect an accuracy of at least $A_{\text{ID}}$ on ID data. We hope that this relation also holds under domain shift. To predict $\widetilde{s}_c(x) = p_{\text{ID}}(c(x) \neq y \mid s_c(x))$ we rely on an architecture that is constrained to be monotonic as proposed in

([Müller et al., 2021](#)). Therefore, we do not change the order of the scores and equip the transformed score with an inductive bias that is consistent with [Criterion 3.1](#). The transformed score also has a predictable extrapolation behavior which is helpful when the distribution shifts. Note that since the transformation is monotonic, a threshold for the original score is also a valid threshold for the transformed score and *vice versa*. Therefore, we can also interpret this approach as estimating an incompetence threshold such that a certain accuracy is achieved.

Accordingly, [Figure 7](#) depicts the ID-Gain and Coverage for ViM and Logit (transformed), if we select $1 - A_{ID}$ as the incompetence threshold. The transformed ViM score suggests that we achieve in most cases at least the ID accuracy, but at the cost of small coverage. The transformed Logit score has higher coverage, but it often fails to reproduce the ID accuracy (e.g., in the TerraIncognita data set). However, while we attain the ID accuracy for most cases, we still observe some failure cases, which makes the approach only tentative. Note, that these results also suggest that the information contained in the logits is not sufficient to give suitable competence regions in the sense of our question.

## 5  Conclusion

Accepting only predictions from the competence region of a classifier increases its accuracy dramatically under domain shift. Determining the fraction of samples where the classifier could be considered competent is a question of how to approach the trade-off between accuracy and coverage. Addressing this trade-off via the incompetence threshold is application-dependent and particularly challenging in the domain generalization (DG) setting where the test distribution differs from the training distribution per definition. Still, we showed that even in DG, it is possible to achieve higher than in-distribution accuracy under domain shift – at the price of potentially diminished coverage (see [Figure 2](#) or [Appendix A.2](#)).

Furthermore, we investigated a coverage-oriented threshold that would reject only a pre-defined fraction (e.g., 5%) of all instances from the training distribution. In this case, we achieved a considerable improvement under distribution shift compared to a naive application where no samples are rejected. However, at this particular threshold, we could recover the ID accuracy only in some settings. Thus, we also studied whether we can learn an accuracy-oriented threshold where some predefined ID accuracy is guaranteed in the competence region. This approach was able to replicate the ID accuracy in the competence region for most investigated domain shifts. However, for a few domains, OOD accuracy drops significantly below the expected ID, calling for a more detailed understanding of the behavior of incompetence scores in DG. Nevertheless, we observed that accuracy in the competence region behaves monotonically with the threshold $\alpha$ (see [Proposition 3.1](#) and [Section 4.2](#)).

Finally, we investigated differences between the closed and open world settings. We found that in the open world setting, feature-based methods, such as Deep-KNN ([Sun et al., 2022](#)) and ViM ([Wang et al., 2022](#)), elicit a particularly useful competence region. In a closed world DG setting, a clear winner does not emerge, but ViM and Deep-KNN seem to be competitive to logit-based approaches. We also analyzed whether we could find differences in the accuracy of the competence region with respect to different classifiers. We could not find statistically significant effects on the accuracy in the competence region, leaving the benefit of robust algorithms for DG and different architectures for enlarged competence regions questionable.

All post-hoc methods investigated in this work are comparably fast to evaluate and therefore easily accessible for practitioners. However, the resolution of the trade-off between accuracy and coverage is not yet satisfactory in all cases, calling for more research on better competence scores. One interesting avenue concerns the use of *multivariate* scores (i.e., a combination of multiple scores) with the potential to elicit better competence regions.

### Acknowledgments

JM and UK were supported by Informatics for Life funded by the Klaus Tschira Foundation. STR and FD were supported by the Deutsche Forschungsgemeinschaft (DFG, German Research Foundation) under Germany's Excellence Strategy EXC-2181/1 - 390900948 (the Heidelberg STRUCTURES Cluster of Excellence). We thank the Center for Information Services and High Performance Computing (ZIH) at TU Dresden for its facilities for high throughput calculations.

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

# A    Appendix

In this Appendix, we describe optimization procedures in detail, give additional detailed results and describe the open world data sets in detail. Furthermore, we give proof for Proposition 3.1.

## A.1    Detailed Qualitative Results

Figure 8 and Figure 9 show for the PACS data set the three images attaining the highest and the lowest incompetence scores per class respectively. Images with lower scores achieve higher accuracy compared to the highest-scored images.

## A.2    Detailed Quantitative Results - Dependence on Competence Threshold

We show the accuracy on OOD test data in dependence on the competence threshold $\alpha$ for the DG data sets PACS, OfficeHome, VLCS, TerraIncognita, DomainNet and SVIRO in Figure 10, Figure 11, Figure 12, Figure 13, Figure 14 and Figure 15 respectively. We only show results for Deep-KNN, Logit, ViM and GMM applied on the ERM classifier. For Deep-KNN, Logit, and Vim we see in almost all cases the monotonic behavior as predicted in Proposition 3.1. On the DG data sets VLCS, DomainNet and TerraIncognita the GMM score fails to show this monotonic behavior for some test domains. Therefore, GMM has not the monotonic behavior we would expect from an admissible incompetence score. For some domains, all scores do not behave as we aimed for. For instance, in the LabelMe test domain in VLCS (see Figure 12) we cannot achieve the ID accuracy for all thresholds $\alpha$ and all incompetence scores. While this behavior is extremely rare in our experiments (for the feature- and logit-based scores), it shows that the current competence scores can fail for some domain shifts.

In Figure 16 we also show results for different thresholds according to their percentile in the ID distribution. We can see that the relative performance of the scores stays considerably stable.

## A.3    Detailed Quantitatively Results – Extensive Study

In Table 2, Table 3 and Table 4 we give detailed results for all DG data sets considered in this work: PACS, VLCS, OfficeHome, TerraIncognita, DomainNet, and SVIRO. We list the accuracies in the competence region where the incompetence threshold is chosen as the 95% percentile of the ID validation set. Here we show the median, the 5% and 95% percentiles over all test domains and classifiers. We can see that the deviations between different test domains are quite severe indicating different strengths of domain shifts across the DG tasks. The main observations in Section 4.4 (e.g. the OOD-gain is quite significant and feature-based methods [ViM; Deep-KNN] are very successfull) hold across the different DG tasks.

## A.4    Open World Setting

**Open World Creation**    We use additional data to extend the closed world data sets to the open world setting. We use similar domains of other data sets with disjunct classes to generalize the DG data sets. The ID data sets and the open world extensions are listed in Table 5. We show examples of test data (closed world) and open world samples for all DG data sets. For PACS (in Figure 21) for VLCS (in Figure 22), for OfficeHome (in Figure 23) and for TerraIncognita (in Figure 24)

**Open World Results**    Figure 17 shows the ID-Gain and OOD-Gain for all incompetence scores considered in this work depending on the fraction of open world samples. We see that ViM and Deep-KNN are particularly able to delineate unknown class instances from known class instances resulting in an improved ID- and OOD-Gain across all open world fractions.

In Figure 18 we investigate the behavior of different scores in detail. It shows the AUROC of delineating ID data vs. correctly classified samples of the test domain, ID data vs. wrongly classified samples of the test domain, and ID data vs. unknown class instances in general. Here we consider an unknown test domain where 25% of all samples are open world outliers. We see an interesting behavior here: ViM and Deep-KNN

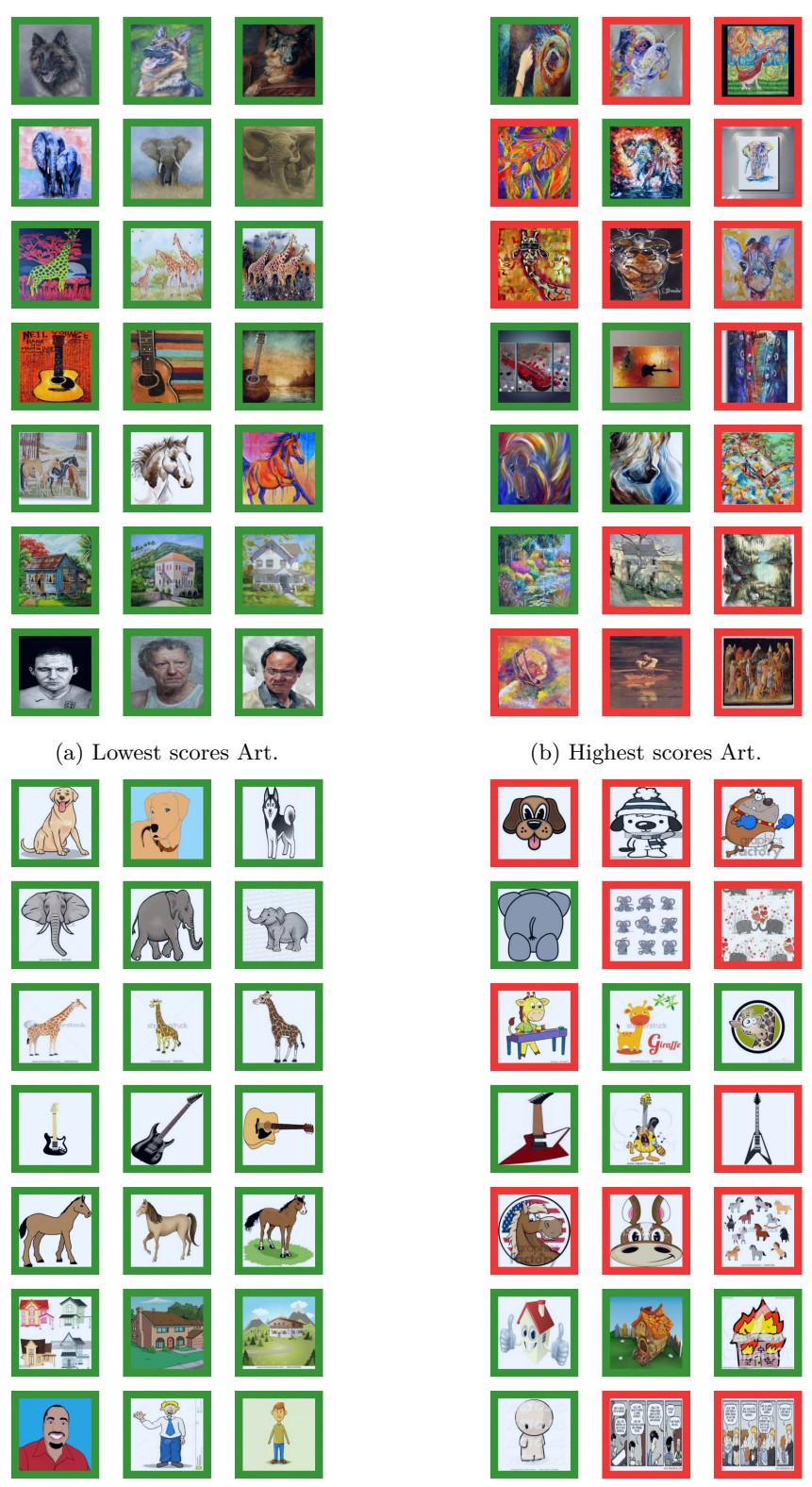

(a) Lowest scores Art.   (b) Highest scores Art.

(c) Lowest scores Cartoon.   (d) Highest scores Cartoon.

Figure 8: Images with highest and lowest scores for different domains on PACS. Scores are computed with Deep-KNN on ERM.

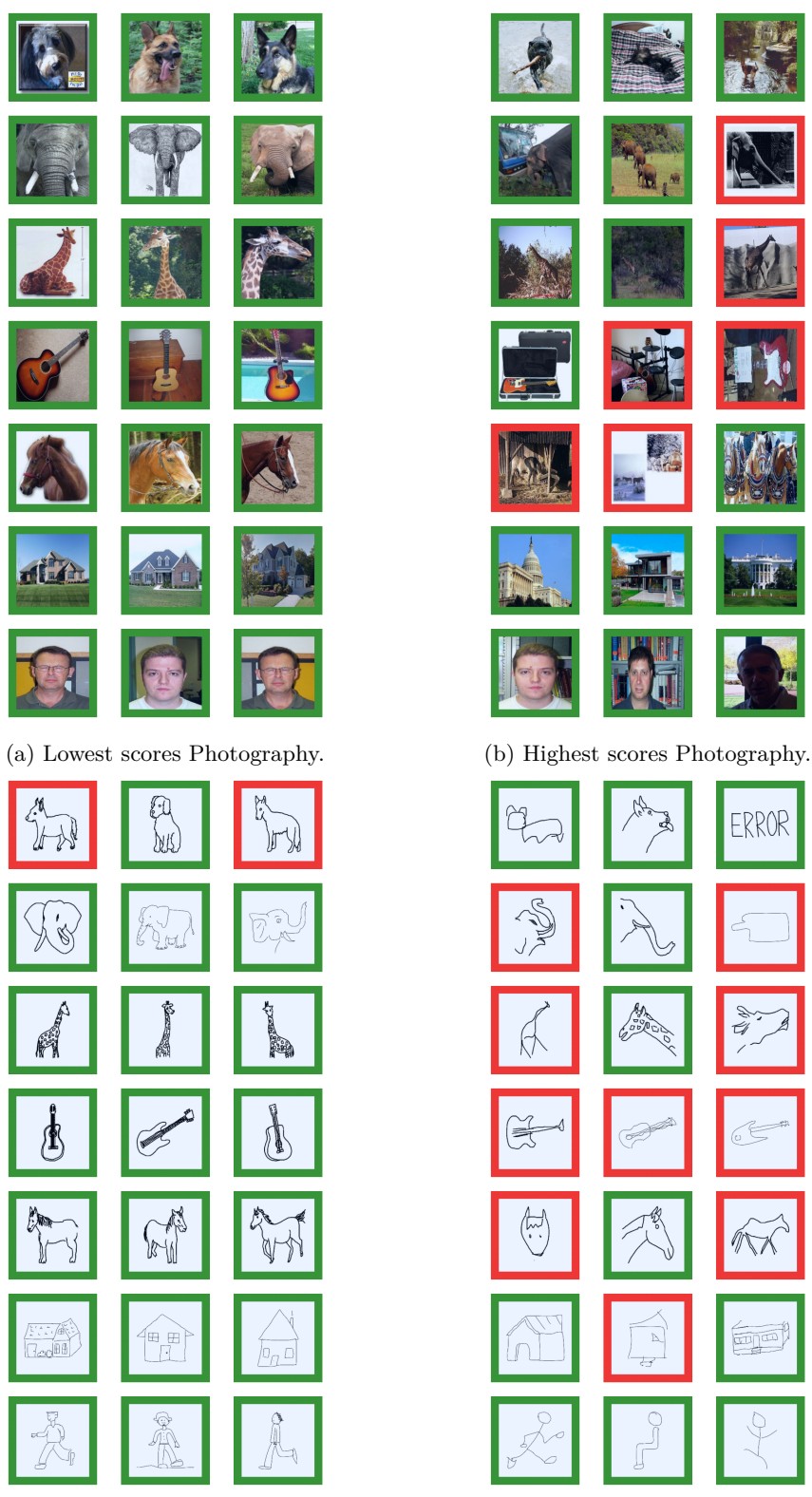

(a) Lowest scores Photography.          (b) Highest scores Photography.

(c) Lowest scores Sketch.          (d) Highest scores Sketch.

Figure 9: Images with highest and lowest scores for different domains on PACS. Scores are computed with Deep-KNN on ERM.

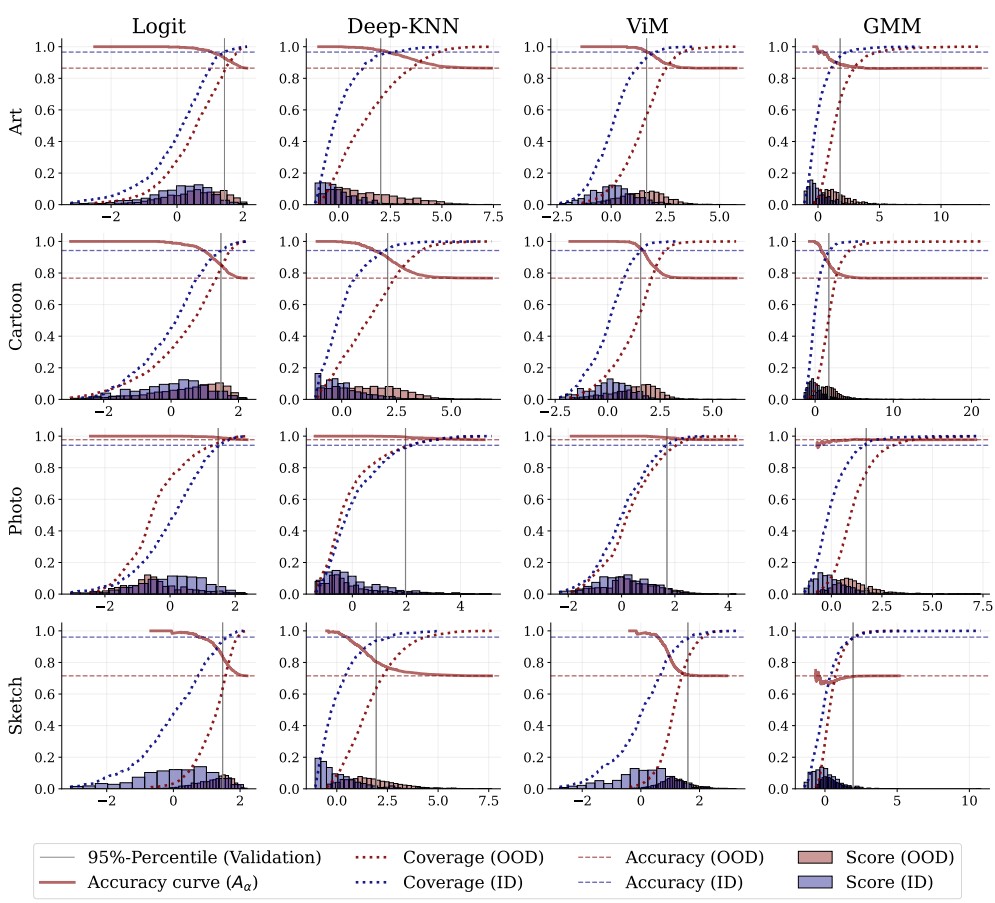

Figure 10: The accuracy of the ERM classifier on OOD data $A_{\text{OOD}}(\alpha)$ as the competence region is enlarged by increasing the allowed incompetence $\alpha$. Here we show the results for all DG tasks of the PACS data set.

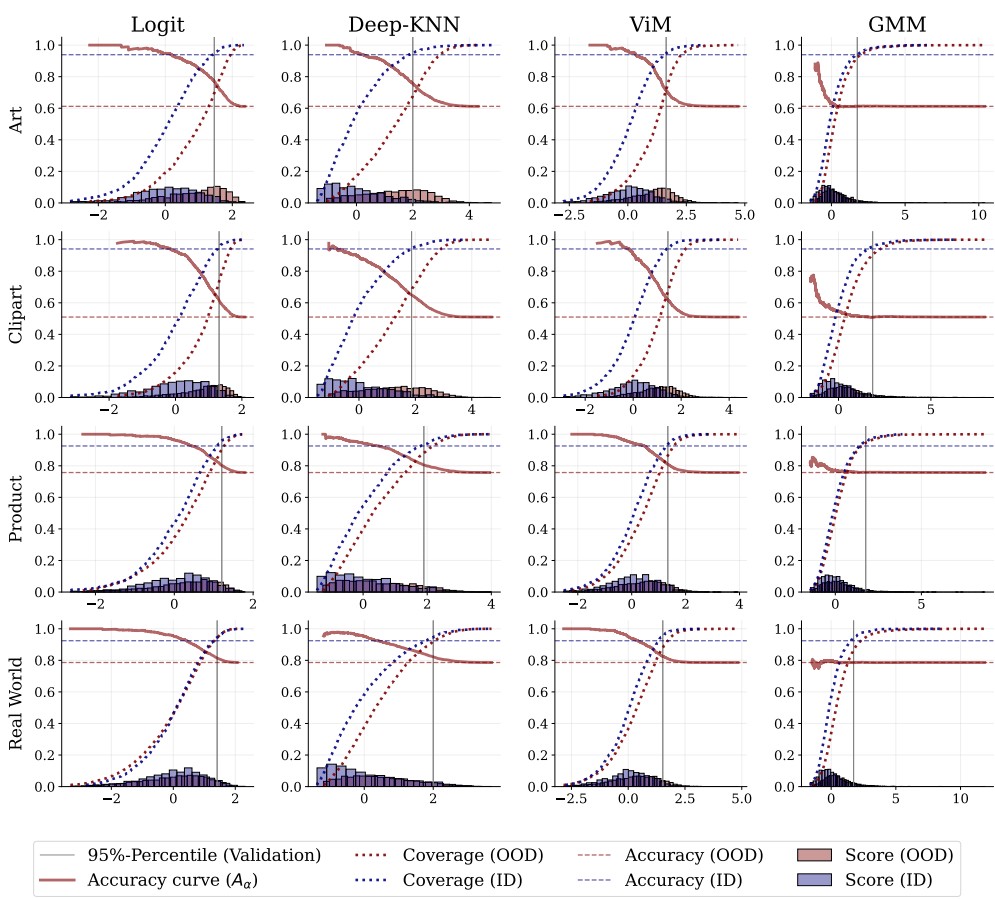

Figure 11: The accuracy of the ERM classifier on OOD data $A_{OOD}(\alpha)$ as the competence region is enlarged by increasing the allowed incompetence $\alpha$. Here we show the results for all DG tasks of the OfficeHome data set.

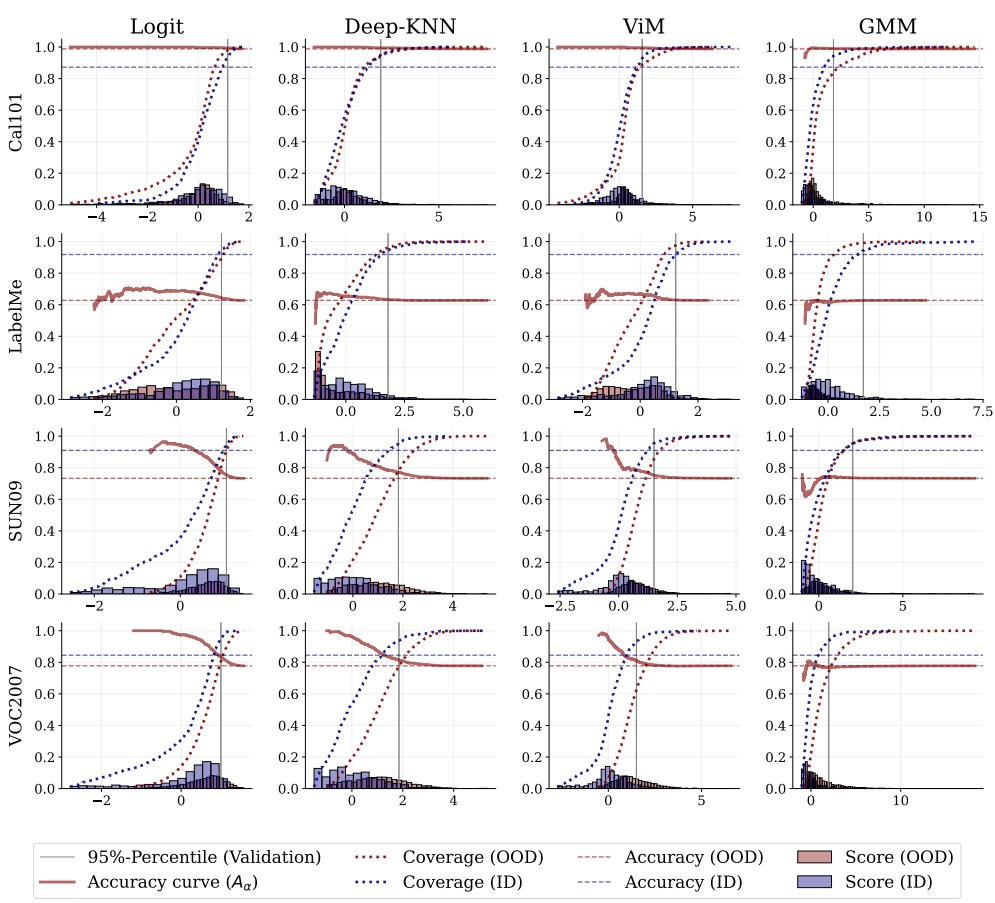

Figure 12: The accuracy of the ERM classifier on OOD data $A_{OOD}(\alpha)$ as the competence region is enlarged by increasing the allowed incompetence $\alpha$. Here we show the results for all DG tasks of the VLCS data set.

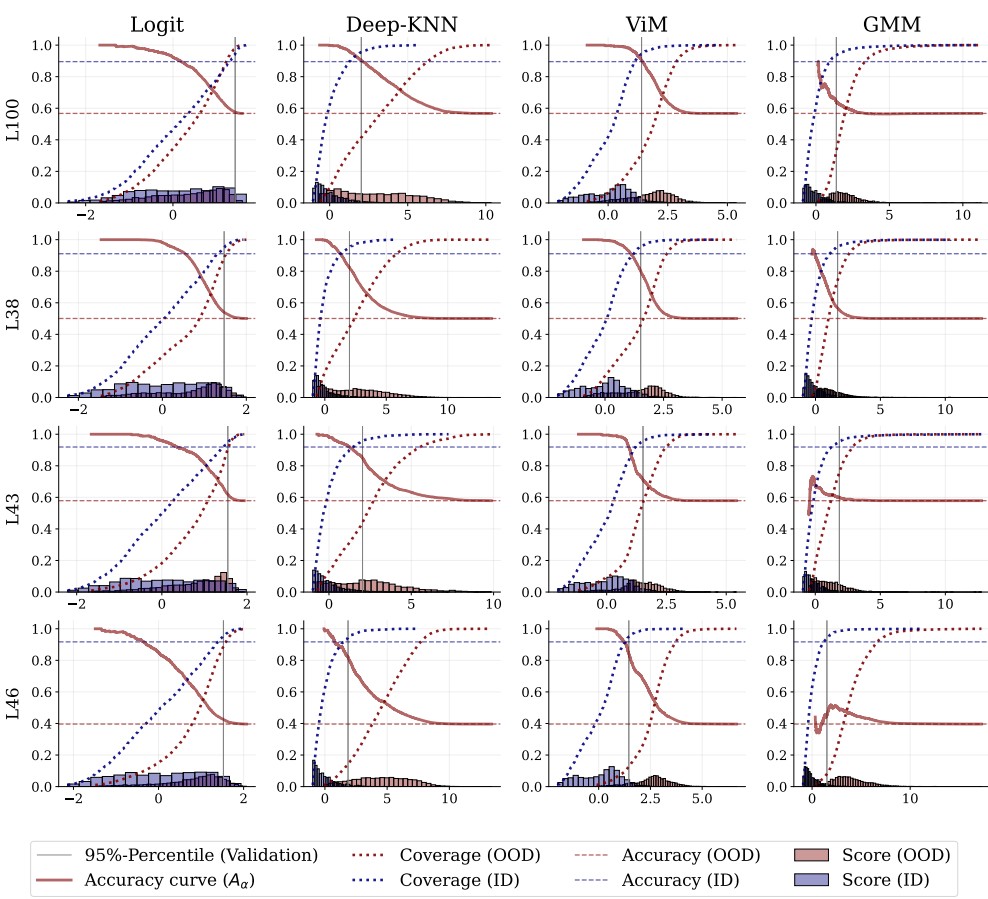

Figure 13: The accuracy of the ERM classifier on OOD data $\mathrm{A_{OOD}}(\alpha)$ as the competence region is enlarged by increasing the allowed incompetence $\alpha$. Here we show the results for all DG tasks of the TerraIncognita data set.

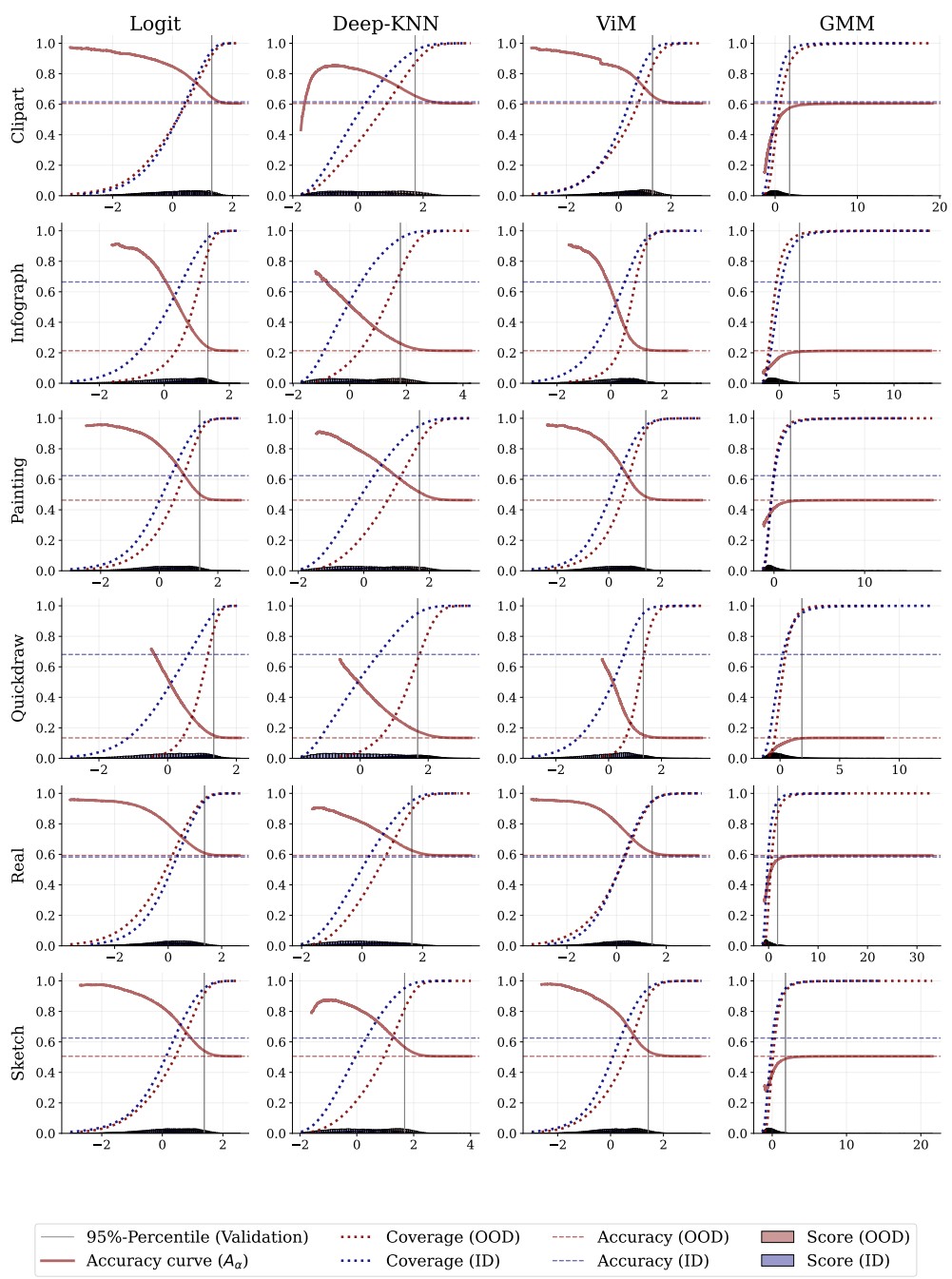

Figure 14: The accuracy of the ERM classifier on OOD data $A_{OOD}(\alpha)$ as the competence region is enlarged by increasing the allowed incompetence $\alpha$. Here we show the results for all DG tasks of the DomainNet data set.

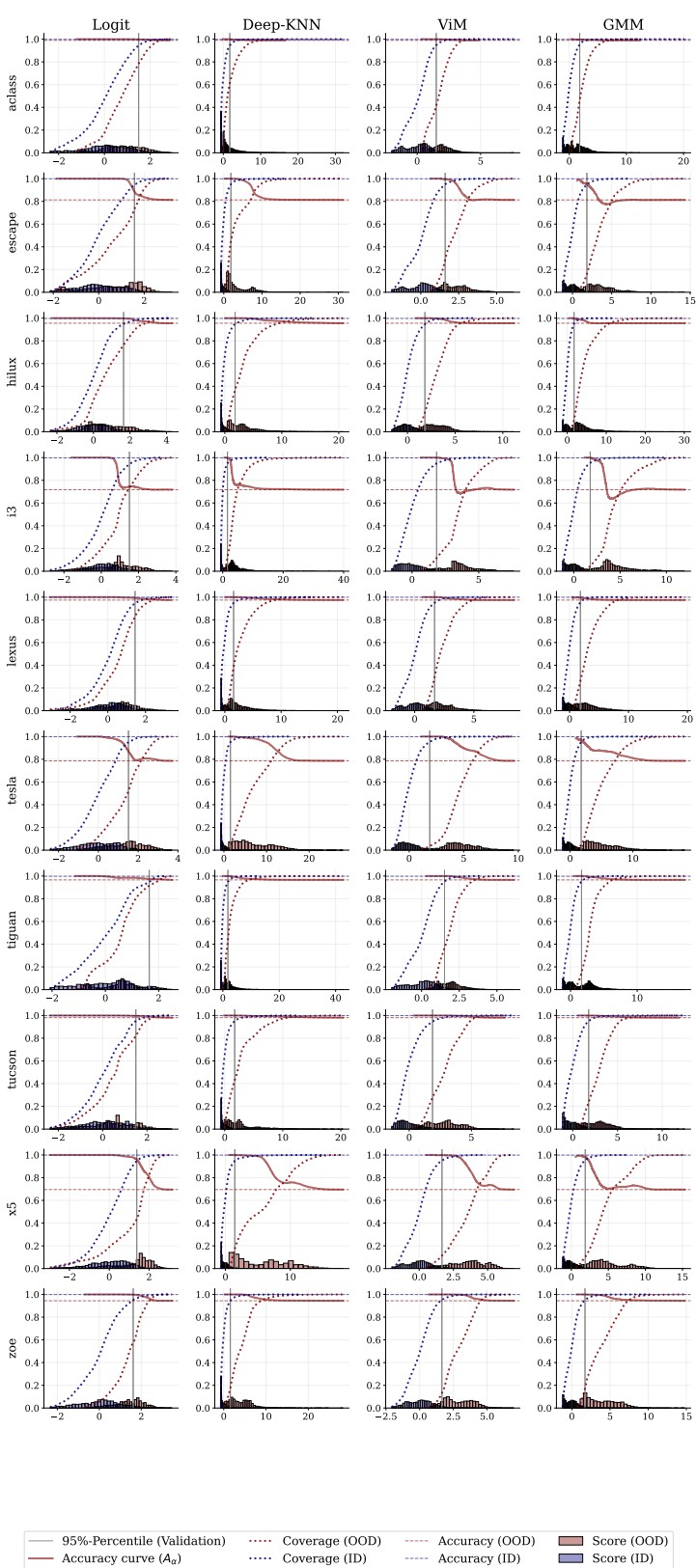

Figure 15: The accuracy of the ERM classifier on OOD data $A_{\text{OOD}}(\alpha)$ as the competence region is enlarged by increasing the allowed incompetence $\alpha$. Here we show the results for all DG tasks of the SVIRO data set.

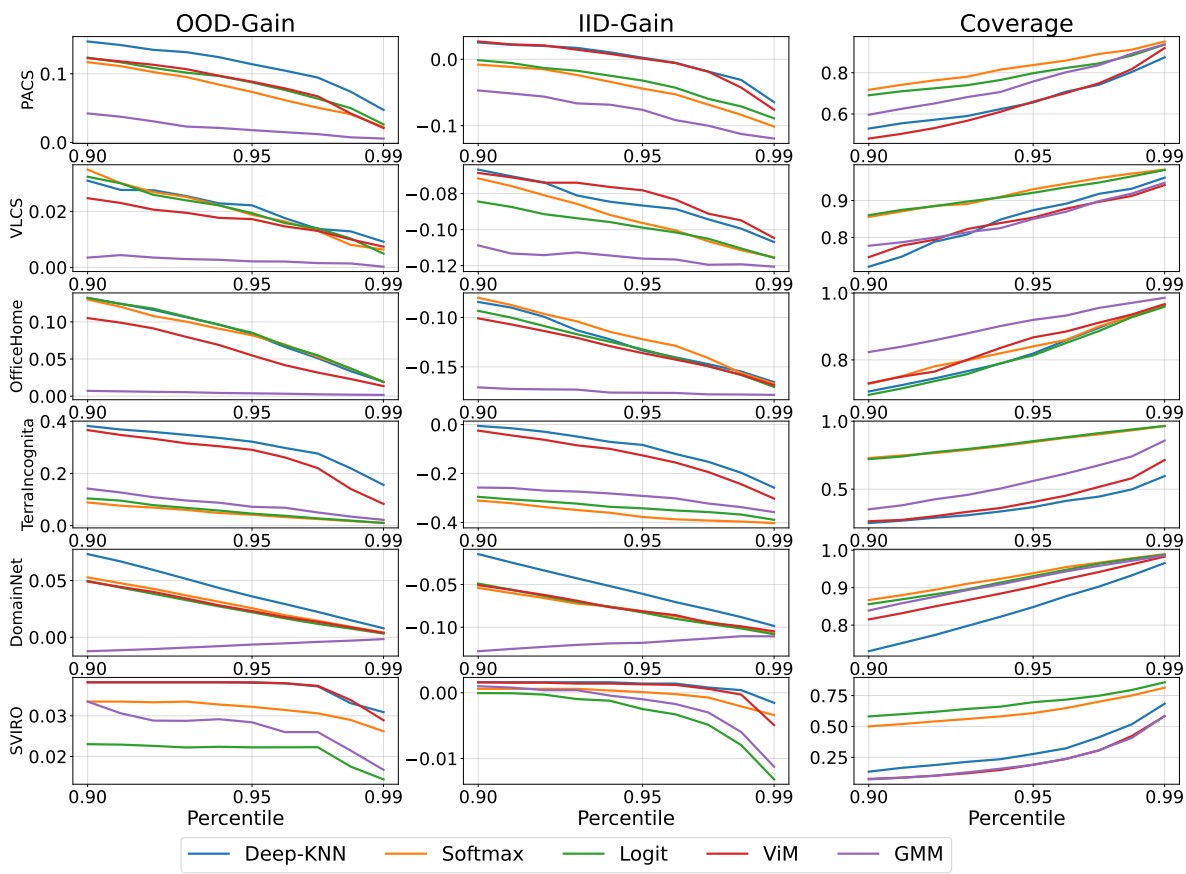

Figure 16: Median of accuracies in competence region for different thresholds (percentiles of ID distribution) over all domain roles and classifiers.

**PACS**

| In Percentages (%) | Art OOD-Gain ↑ | Art ID-Gap ↑ | Art Coverage ↑ | Cartoon OOD-Gain ↑ | Cartoon ID-Gap ↑ | Cartoon Coverage ↑ | Painting OOD-Gain ↑ | Painting ID-Gap ↑ | Painting Coverage ↑ | Sketch OOD-Gain ↑ | Sketch ID-Gap ↑ | Sketch Coverage ↑ |
|---|---|---|---|---|---|---|---|---|---|---|---|---|
| Deep-KNN | **12** [8-14] | **2** [-1-3] | 66 [57-72] | 14 [11-18] | -1 [-6-1] | 63 [56-71] | **2** [1-2] | **3** [3-5] | 94 [93-96] | **13** [9-17] | -9 [-15-5] | 59 [53-70] |
| ViM | 11 [7-14] | 1 [-2-3] | 56 [53-71] | **17** [11-19] | **0** [-4-2] | 55 [49-60] | **2** [1-2] | **3** [3-5] | 91 [86-95] | 3 [1-21] | -15 [-22-5] | 77 [35-86] |
| Softmax | 7 [6-10] | -4 [-5-1] | 86 [77-88] | 7 [5-10] | -10 [-12-4] | 82 [78-86] | 1 [1-2] | **3** [2-5] | 97 [96-98] | 12 [7-15] | **-8** [-19-5] | 68 [63-82] |
| Logit | 9 [6-11] | -3 [-4-1] | 82 [65-85] | 10 [7-11] | -9 [-10-2] | 77 [72-84] | **2** [1-2] | **3** [3-5] | 95 [93-97] | 12 [4-16] | **-8** [-22-5] | 65 [57-90] |
| Energy | 9 [6-11] | -3 [-4-1] | 81 [64-84] | 9 [7-11] | -9 [-10-2] | 77 [72-84] | **2** [1-2] | **3** [3-5] | 95 [92-97] | 11 [3-16] | **-8** [-23-4] | 66 [56-91] |
| Energy-React | 9 [6-11] | -3 [-4-1] | 80 [63-84] | 9 [7-11] | -8 [-10-2] | 77 [71-84] | **2** [1-2] | **3** [3-5] | 95 [92-97] | 11 [3-16] | **-8** [-23-4] | 66 [56-91] |
| Mahalanobis | 2 [0-10] | -7 [-11-1] | 67 [56-94] | 7 [0-12] | -10 [-14-5] | 62 [50-94] | 0 [0-1] | **3** [2-4] | 87 [78-96] | 0 [-1-17] | -19 [-24-9] | 89 [28-97] |
| GMM | 3 [1-10] | -7 [-11-2] | 66 [54-86] | 8 [4-13] | -9 [-15-2] | 56 [50-67] | 1 [0-1] | **3** [2-4] | 83 [76-94] | 0 [-1-17] | -19 [-24-9] | 87 [28-97] |
| PCA | 0 [-1-7] | -11 [-14-0] | 79 [63-93] | 3 [1-11] | -13 [-17-4] | 72 [56-81] | 0 [-1-1] | **3** [0-4] | 82 [70-94] | 0 [-1-13] | -18 [-24-12] | 95 [28-100] |

**VLCS**

| | Cal101 OOD-Gain ↑ | Cal101 ID-Gap ↑ | Cal101 Coverage ↑ | Label OOD-Gain ↑ | Label ID-Gap ↑ | Label Coverage ↑ | SUN09 OOD-Gain ↑ | SUN09 ID-Gap ↑ | SUN09 Coverage ↑ | VOC2007 OOD-Gain ↑ | VOC2007 ID-Gap ↑ | VOC2007 Coverage ↑ |
|---|---|---|---|---|---|---|---|---|---|---|---|---|
| Deep-KNN | **2** [1-4] | 13 [11-15] | 93 [89-96] | 0 [0-1] | -27 [-28-22] | 98 [96-99] | 2 [1-5] | **-15** [-19-6] | 79 [73-85] | **4** [4-6] | **-6** [-9-2] | 76 [66-81] |
| ViM | **2** [0-3] | **14** [10-15] | 87 [68-88] | 0 [0-0] | -27 [-28-22] | 98 [97-99] | 2 [1-4] | -16 [-20-5] | 85 [75-88] | **4** [2-6] | -7 [-9-3] | 68 [54-74] |
| Softmax | 1 [0-1] | 12 [9-14] | 98 [97-99] | **1** [1-2] | **-26** [-28-21] | 94 [92-96] | **3** [2-4] | **-15** [-19-6] | 89 [85-92] | 3 [2-5] | **-6** [-11-3] | 90 [85-93] |
| Logit | 1 [0-2] | 13 [9-14] | 98 [95-99] | **1** [0-1] | **-26** [-28-22] | 96 [90-97] | **3** [2-4] | **-15** [-19-6] | 87 [84-92] | 3 [2-5] | **-6** [-10-2] | 89 [81-94] |
| Energy | 1 [0-2] | 12 [9-14] | 98 [95-99] | 0 [0-1] | **-26** [-28-22] | 96 [90-98] | 2 [2-3] | **-15** [-19-6] | 89 [85-92] | 3 [2-5] | **-6** [-11-2] | 88 [79-94] |
| Energy-React | 1 [0-2] | 12 [9-14] | 98 [95-99] | 0 [0-1] | **-26** [-28-22] | 96 [90-98] | 2 [2-3] | **-15** [-19-6] | 89 [85-92] | 3 [2-5] | **-6** [-11-2] | 88 [79-93] |
| Mahalanobis | 1 [0-3] | 13 [9-15] | 84 [76-95] | 0 [0-0] | -27 [-28-22] | 98 [94-99] | 0 [0-1] | -17 [-22-8] | 95 [87-96] | 0 [-1-3] | -9 [-12-6] | 76 [70-94] |
| GMM | 1 [0-3] | 13 [9-15] | 81 [49-86] | 0 [0-0] | -27 [-28-23] | 98 [98-99] | 0 [0-2] | -17 [-22-7] | 95 [77-96] | 0 [-2-3] | -9 [-12-6] | 74 [52-80] |
| PCA | 1 [0-3] | 13 [8-15] | 82 [69-90] | 0 [0-0] | -27 [-28-22] | 99 [98-99] | 0 [0-1] | -17 [-22-8] | 95 [85-97] | 0 [-2-3] | -9 [-13-6] | 77 [56-83] |

**OfficeHome**

| | Art OOD-Gain ↑ | Art ID-Gap ↑ | Art Coverage ↑ | Clipart OOD-Gain ↑ | Clipart ID-Gap ↑ | Clipart Coverage ↑ | Product OOD-Gain ↑ | Product ID-Gap ↑ | Product Coverage ↑ | Real World OOD-Gain ↑ | Real World ID-Gap ↑ | Real World Coverage ↑ |
|---|---|---|---|---|---|---|---|---|---|---|---|---|
| Deep-KNN | **13** [11-17] | **-15** [-21-5] | 72 [65-77] | **14** [9-16] | **-28** [-31-15] | 69 [62-78] | 5 [3-7] | -12 [-13-2] | 89 [85-94] | **4** [2-6] | -10 [-14-1] | 92 [87-94] |
| ViM | 8 [6-12] | -21 [-27-4] | 76 [69-80] | 11 [6-15] | **-28** [-33-19] | 68 [56-83] | 4 [2-5] | -12 [-14-1] | 90 [87-95] | **4** [2-4] | -11 [-13-1] | 89 [87-94] |
| Softmax | 11 [8-16] | -17 [-23-8] | 76 [66-86] | 10 [8-16] | **-28** [-33-18] | 75 [64-82] | **7** [3-8] | **-11** [-12-2] | 87 [84-95] | **4** [3-8] | **-8** [-12-1] | 93 [84-95] |
| Logit | **13** [9-17] | -16 [-21-6] | 73 [66-82] | 11 [8-16] | **-28** [-35-15] | 74 [63-81] | 5 [2-8] | -12 [-13-2] | 90 [83-95] | 3 [2-8] | -9 [-14-1] | 94 [83-96] |
| Energy | 12 [9-16] | -17 [-22-6] | 75 [66-82] | 10 [6-16] | -30 [-36-16] | 74 [64-83] | 4 [2-7] | -12 [-14-2] | 91 [84-96] | 3 [2-7] | -10 [-14-1] | 95 [83-96] |
| Energy-React | 12 [9-16] | -17 [-22-6] | 74 [65-82] | 11 [7-16] | **-28** [-36-16] | 74 [63-82] | 5 [2-7] | -12 [-14-2] | 91 [84-95] | 3 [2-7] | -10 [-14-1] | 95 [84-96] |
| Mahalanobis | 1 [0-12] | -28 [-33-4] | 91 [70-93] | 1 [0-6] | -40 [-43-21] | 89 [75-92] | 0 [0-2] | -16 [-18-1] | 94 [93-95] | 0 [0-2] | -15 [-17-1] | 91 [87-95] |
| GMM | 1 [0-11] | -28 [-34-5] | 92 [69-93] | 0 [0-6] | -40 [-43-22] | 90 [77-92] | 0 [0-2] | -16 [-18-1] | 95 [93-95] | 0 [0-2] | -15 [-17-1] | 91 [87-95] |
| PCA | 0 [0-8] | -28 [-34-8] | 92 [77-95] | 0 [-1-6] | -40 [-43-22] | 92 [77-94] | 0 [0-2] | -17 [-19-1] | 95 [94-96] | 0 [0-2] | -15 [-18-0] | 92 [89-96] |

**TerraIncognita**

| | L100 OOD-Gain ↑ | L100 ID-Gap ↑ | L100 Coverage ↑ | L38 OOD-Gain ↑ | L38 ID-Gap ↑ | L38 Coverage ↑ | L43 OOD-Gain ↑ | L43 ID-Gap ↑ | L43 Coverage ↑ | L46 OOD-Gain ↑ | L46 ID-Gap ↑ | L46 Coverage ↑ |
|---|---|---|---|---|---|---|---|---|---|---|---|---|
| Deep-KNN | **34** [30-45] | **1** [-18-5] | 31 [16-41] | **30** [18-38] | **-16** [-36-6] | 45 [34-53] | **23** [11-34] | **-11** [-20-1] | 44 [34-53] | 42 [2-54] | **-8** [-56-4] | 14 [10-30] |
| ViM | **34** [25-38] | -6 [-13-4] | 32 [12-47] | 23 [14-36] | -19 [-41-11] | 51 [33-63] | 17 [8-22] | -16 [-25-12] | 47 [41-57] | **45** [20-55] | **-7** [-34-6] | 13 [4-34] |
| Softmax | 4 [2-12] | -31 [-46-18] | 84 [74-95] | 6 [1-9] | -43 [-55-30] | 83 [72-94] | 3 [2-7] | -28 [-34-23] | 91 [80-96] | 4 [1-16] | -48 [-60-34] | 78 [55-93] |
| Logit | 5 [2-20] | -30 [-45-12] | 85 [56-96] | 9 [2-16] | -39 [-54-23] | 76 [65-93] | 2 [1-6] | -30 [-33-25] | 94 [82-98] | 3 [1-19] | -47 [-61-31] | 86 [50-94] |
| Energy | 5 [1-21] | -30 [-45-11] | 86 [49-97] | 10 [2-17] | -38 [-54-21] | 75 [62-91] | 2 [0-6] | -31 [-33-25] | 94 [83-99] | 3 [1-20] | -46 [-61-28] | 84 [49-94] |
| Energy-React | 5 [2-21] | -30 [-45-12] | 86 [50-97] | 10 [2-17] | -38 [-54-21] | 75 [62-91] | 2 [0-6] | -31 [-33-25] | 94 [83-99] | 3 [1-21] | -46 [-61-28] | 84 [49-94] |
| Mahalanobis | 6 [-2-24] | -26 [-48-14] | 62 [23-91] | 5 [1-20] | -38 [-55-28] | 75 [57-94] | 3 [0-12] | -28 [-36-19] | 62 [37-90] | 13 [1-44] | -35 [-56-7] | 15 [3-92] |
| GMM | 10 [-3-27] | -25 [-45-9] | 33 [19-77] | 7 [1-28] | -37 [-49-21] | 70 [45-89] | 5 [0-12] | -28 [-34-19] | 62 [37-72] | 13 [3-44] | -34 [-54-7] | 14 [3-54] |
| PCA | 0 [-2-11] | -36 [-48-20] | 88 [50-99] | 1 [0-18] | -44 [-53-28] | 96 [57-99] | 0 [-1-6] | -33 [-36-25] | 90 [60-96] | 4 [0-26] | -45 [-56-28] | 72 [17-82] |

Table 2: Accuracy on competence region of OOD domain for different *PACS*, *OfficeHome*, *VLCS* and *TerraIncognita* domains and incompetence scores. As the threshold for the competence regions, we choose the 95% percentile of the ID validation set. For all metrics, a higher value means better performance (↑). All displayed values are medians over different domain roles and classifiers, brackets indicate 90% confidence interval.

are well-able to filter out wrongly classified samples, but also filter out many correctly classified samples. The logit-based scores (Logit, Softmax, Energy, Energy-React) are less successful in filtering out wrongly classified samples, but also keep more correctly classified samples. In the optimal case, we would expect that the AUROC of ID vs. correct test data is ≤ 0.5 and the AUROC of ID vs. false OOD data is 1. This would imply that we could successfully filter out wrongly predicted samples and keep a high coverage. Figure 18 shows that ViM and Deep-KNN are capable of filtering out new class instances across all DG data sets. For all other scores, we can find data sets where this behavior is not achieved. Consequently, ViM and Deep-KNN work best when unknown class instances occur.

## A.5 Training Details and Classifiers

All classifiers are trained using the DomainBed repository [2]. We train three different neural network architectures with Emprirical-Risk-Minimization, shortly ERM (Vapnik, 1999). Namely, a ResNet based architecture (He et al., 2016), a Vision Transformer (Dosovitskiy et al., 2020) and a Swin Transformer (Liu et al., 2021). If we just refer to ERM, we mean the ResNet-based architecture. Furthermore, we train classifiers with various recent DG algorithms, namely Fish (Shi et al., 2021), GroupDRO (Sagawa et al., 2019), SD (Pezeshki et al., 2021), SagNet (Nam et al., 2021), Mixup (Yan et al., 2020) and VREx (Krueger et al., 2021).

We use all the standard settings provided in the DomainBed repository and train all classifiers with hyperparameters proposed in the repository. The Vision Transformer and SwinTransformer are trained with hyperparameters found useful on these data sets and architectures as in (Wenzel et al., 2022). Each model is trained for 100 epochs on the smaller data sets (PACS, VLCS, TerraIncognita and OfficeHome) and for

---

[2] https://github.com/facebookresearch/DomainBed

| In Percentages (%) | clip | | | info | | | paint | | |
|---|---|---|---|---|---|---|---|---|---|
| | OOD-Gain ↑ | ID-Gap ↑ | Frac ↑ | OOD-Gain ↑ | ID-Gap ↑ | Frac ↑ | OOD-Gain ↑ | ID-Gap ↑ | Frac ↑ |
| Deep-KNN | **4** [2-5] | **4** [2-7] | 86 [82-90] | 3 [0-6] | **-40** [-45−31] | 81 [71-89] | **4** [1-6] | **-9** [-11−6] | 86 [84-92] |
| ViM | **4** [2-6] | **4** [1-6] | 90 [85-93] | 1 [0-9] | -42 [-44−28] | 92 [66-97] | 2 [1-8] | -10 [-13−3] | 92 [80-95] |
| Softmax | 3 [1-3] | 3 [1-5] | 95 [94-97] | **5** [1-7] | **-40** [-42−28] | 78 [74-88] | 3 [1-4] | -10 [-12−8] | 93 [91-95] |
| Logit | 3 [1-4] | **4** [2-5] | 92 [90-96] | 2 [0-8] | -42 [-44−30] | 87 [72-96] | 3 [1-4] | -10 [-12−8] | 93 [90-95] |
| Energy | 3 [1-5] | 3 [1-5] | 91 [88-96] | 1 [0-8] | -42 [-44−30] | 92 [73-97] | 3 [1-4] | -10 [-13−7] | 93 [90-95] |
| Energy-React | 3 [1-5] | 3 [1-5] | 92 [88-96] | 1 [0-8] | -42 [-44−30] | 92 [72-98] | 2 [1-4] | -10 [-13−7] | 93 [91-96] |
| Mahalonobis | -1 [-3-3] | -2 [-3-5] | 86 [84-93] | 0 [-1-4] | -44 [-45−32] | 96 [79-97] | 0 [-1-8] | -14 [-16−3] | 95 [80-97] |
| GMM | -2 [-3-2] | -2 [-4-4] | 87 [84-94] | 0 [-1-3] | -45 [-46−32] | 97 [83-98] | -1 [-1-6] | -14 [-16−4] | 95 [80-97] |
| PCA | -2 [-3-1] | -2 [-4-3] | 86 [84-93] | 0 [-1-1] | -45 [-47−32] | 96 [87-98] | -1 [-1-1] | -14 [-16−10] | 96 [90-97] |

| | quick | | | real | | | sketch | | |
|---|---|---|---|---|---|---|---|---|---|
| | OOD-Gain ↑ | ID-Gap ↑ | Coverage ↑ | OOD-Gain ↑ | ID-Gap ↑ | Coverage ↑ | OOD-Gain ↑ | ID-Gap ↑ | Coverage ↑ |
| Deep-KNN | **3** [0-4] | **-48** [-52−27] | 78 [65-91] | **3** [1-3] | **5** [2-9] | 92 [89-95] | **6** [2-9] | **-4** [-6-0] | 83 [79-87] |
| ViM | 2 [0-3] | -49 [-54−27] | 80 [65-99] | 2 [0-4] | 4 [2-11] | 94 [88-97] | 3 [0-6] | -7 [-9−1] | 90 [86-95] |
| Softmax | 0 [0-2] | -51 [-54−27] | 94 [88-98] | 1 [1-2] | 3 [1-8] | 97 [97-98] | 3 [2-4] | -7 [-9−1] | 93 [92-94] |
| Logit | 1 [0-2] | -50 [-54−27] | 89 [72-95] | 2 [1-2] | 3 [2-7] | 97 [96-98] | 3 [1-5] | -7 [-8−1] | 93 [91-94] |
| Energy | 1 [0-2] | -50 [-54−27] | 88 [71-98] | 1 [0-2] | 3 [1-7] | 98 [95-98] | 3 [0-4] | -7 [-9−2] | 94 [91-95] |
| Energy-React | 1 [0-2] | -50 [-54−27] | 88 [72-98] | 1 [0-2] | 3 [1-7] | 98 [95-98] | 3 [0-4] | -8 [-9−2] | 94 [91-95] |
| Mahalonobis | 0 [-1-1] | -52 [-55−27] | 94 [68-100] | -1 [-3-5] | 1 [-2-11] | 87 [81-91] | -1 [-1-5] | -10 [-13−2] | 93 [86-95] |
| GMM | 0 [-1-0] | -52 [-55−27] | 95 [75-100] | -1 [-3-4] | 0 [-3-10] | 87 [79-90] | -1 [-2-3] | -11 [-14−2] | 93 [88-95] |
| PCA | 0 [-1-0] | -52 [-55−27] | 94 [75-99] | -2 [-3-3] | 1 [-2-9] | 87 [79-90] | -1 [-2-2] | -11 [-13−3] | 93 [89-95] |

Table 3: Accuracy on competence region of OOD domain for different DomainNet domains and incompetence scores. As the threshold for the competence regions, we choose the 95% percentile of the ID validation set. For all metrics, a higher value means better performance (↑). All displayed values are medians over different domain roles and classifiers, brackets indicate 90% confidence interval.

| In Percentages (%) | aclass | | | escape | | | hilux | | | i3 | | | lexu | | |
|---|---|---|---|---|---|---|---|---|---|---|---|---|---|---|---|
| | OOD-Gain ↑ | ID-Gap ↑ | Coverage ↑ | OOD-Gain ↑ | ID-Gap ↑ | Coverage ↑ | OOD-Gain ↑ | ID-Gap ↑ | Coverage ↑ | OOD-Gain ↑ | ID-Gap ↑ | Coverage ↑ | OOD-Gain ↑ | ID-Gap ↑ | Frac ↑ |
| Deep-KNN | **2** [1-4] | 0 [0-0] | 56 [19-64] | **14** [1-20] | 0 [-1-0] | 41 [23-63] | **3** [1-8] | 0 [-1-0] | 25 [19-56] | **17** [4-27] | 0 [-3-0] | 15 [10-26] | **4** [1-9] | 0 [0-0] | 27 [14-45] |
| ViM | **2** [1-4] | 0 [0-0] | 42 [26-65] | 11 [1-20] | 0 [-3-0] | 28 [12-56] | **3** [1-7] | 0 [0-0] | 17 [11-33] | **17** [4-28] | 0 [0-0] | 12 [8-17] | **4** [2-9] | 0 [0-0] | 20 [6-44] |
| Softmax | **2** [1-3] | 0 [-1-0] | 82 [71-86] | 11 [1-18] | 0 [-4-0] | 67 [55-74] | 2 [1-7] | 0 [0-0] | 69 [57-72] | 7 [-1-18] | -1 [-30-0] | 51 [33-64] | 3 [1-9] | 0 [-1-0] | 56 [35-77] |
| Logit | 1 [0-3] | 0 [-1-0] | 81 [72-90] | 7 [0-14] | 0 [-15-0] | 73 [65-82] | 2 [1-5] | 0 [-5-0] | 73 [54-80] | 4 [-2-18] | -2 [-30-0] | 54 [35-79] | 2 [1-8] | 0 [-3-0] | 60 [36-81] |
| Energy | 1 [0-3] | 0 [-1-0] | 81 [72-90] | 5 [-1-14] | 0 [-16-0] | 73 [66-85] | 2 [0-5] | 0 [-6-0] | 73 [54-81] | 4 [-2-18] | -2 [-30-0] | 55 [35-79] | 2 [1-7] | 0 [-3-0] | 60 [36-81] |
| Energy-React | 1 [0-3] | 0 [-1-0] | 81 [72-90] | 5 [-1-14] | 0 [-16-0] | 73 [66-85] | 2 [0-5] | 0 [-6-0] | 73 [53-81] | 4 [-2-17] | -2 [-30-0] | 55 [35-79] | 2 [1-8] | 0 [-3-0] | 61 [34-81] |
| Mahalonobis | 1 [0-3] | 0 [-1-0] | 44 [23-94] | 4 [-8-19] | -2 [-20-0] | 19 [17-94] | 2 [-1-7] | 0 [-5-0] | 16 [8-95] | 4 [-17-28] | 0 [-32-0] | 18 [6-93] | 3 [0-9] | 0 [-4-0] | 20 [7-95] |
| GMM | 1 [1-3] | 0 [-1-0] | 44 [24-70] | 3 [-8-19] | -1 [-20-0] | 19 [17-61] | 2 [1-7] | 0 [-4-0] | 16 [8-42] | 7 [-16-28] | 0 [-30-0] | 15 [6-19] | 2 [1-9] | 0 [-4-0] | 21 [7-47] |
| PCA | 0 [0-1] | | 89 [68-99] | -1 [-2-7] | 0 [-3-0] | 83 [67-93] | 1 [0-6] | -1 [-4-0] | 84 [42-94] | 0 [-2-3] | 0 [-30-0] | 90 [48-100] | 0 [0-2] | -3 [-9-1] | 88 [66-95] |

| | tesla | | | tiguan | | | tucson | | | x5 | | | zoe | | |
|---|---|---|---|---|---|---|---|---|---|---|---|---|---|---|---|
| | OOD-Gain ↑ | ID-Gap ↑ | Coverage ↑ | OOD-Gain ↑ | ID-Gap ↑ | Coverage ↑ | OOD-Gain ↑ | ID-Gap ↑ | Coverage ↑ | OOD-Gain ↑ | ID-Gap ↑ | Coverage ↑ | OOD-Gain ↑ | ID-Gap ↑ | Coverage ↑ |
| Deep-KNN | **21** [2-36] | 0 [-1-0] | 5 [2-18] | **1** [1-4] | 0 [0-0] | 43 [22-66] | **2** [0-4] | 0 [0-0] | 37 [19-66] | **20** [13-32] | 0 [0-0] | 16 [5-41] | **13** [2-25] | 0 [-1-0] | 28 [18-46] |
| ViM | **21** [2-37] | 0 [0-0] | 7 [2-17] | **1** [1-4] | 0 [0-0] | 38 [19-67] | **2** [0-4] | 0 [0-0] | **20** [14-65] | 20 [13-31] | 0 [0-0] | 11 [5-23] | **13** [2-25] | 0 [0-0] | 19 [9-43] |
| Softmax | 17 [-2-27] | -1 [-28-0] | 35 [22-45] | **1** [0-3] | 0 [0-0] | 68 [52-84] | **2** [0-4] | 0 [0-0] | 63 [39-81] | **20** [13-30] | -1 [-1-0] | 42 [26-56] | 9 [0-25] | 0 [-12-0] | 55 [36-70] |
| Logit | 4 [-9-21] | -2 [-46-0] | 60 [26-72] | **1** [1-2] | 0 [-2-0] | 79 [56-90] | **2** [0-4] | 0 [-1-0] | 73 [52-88] | 19 [9-28] | -2 [-8-0] | 54 [41-66] | 9 [-1-21] | 0 [-16-0] | 57 [43-71] |
| Energy | 4 [-9-20] | -2 [-46-0] | 60 [26-74] | **1** [0-2] | 0 [-2-0] | 79 [57-90] | **2** [0-4] | 0 [-2-0] | 74 [59-88] | 19 [1-28] | -2 [-17-0] | 56 [42-66] | 5 [-1-21] | 0 [-22-0] | 57 [50-71] |
| Energy-React | 4 [-11-20] | -2 [-48-0] | 60 [27-74] | **1** [0-2] | 0 [-2-0] | 79 [56-91] | **2** [0-4] | 0 [-2-0] | 74 [59-88] | 19 [1-27] | -3 [-17-0] | 56 [45-66] | 5 [-2-21] | 0 [-23-0] | 57 [48-72] |
| Mahalonobis | 20 [0-37] | -1 [-4-0] | 7 [2-94] | **1** [0-3] | 0 [-1-0] | 35 [19-95] | **2** [0-4] | 0 [-2-0] | 23 [9-95] | 18 [-14-31] | -6 [-33-0] | 12 [6-95] | 6 [-4-21] | 0 [-24-0] | 25 [10-93] |
| GMM | 19 [2-37] | 0 [-4-0] | 7 [3-22] | **1** [1-3] | 0 [0-0] | 35 [19-71] | **2** [0-4] | 0 [0-0] | 22 [9-69] | 18 [-15-31] | 0 [-32-0] | 13 [6-24] | 9 [-6-25] | 0 [-14-0] | 26 [10-39] |
| PCA | 2 [0-32] | -6 [-25−1] | 83 [32-91] | 0 [0-2] | -1 [-3-0] | 90 [70-97] | 1 [0-4] | 0 [-3-0] | 90 [66-96] | 7 [-2-17] | -12 [-30−1] | 80 [48-94] | 0 [0-19] | -5 [-21-0] | 91 [57-100] |

Table 4: Accuracy on competence region of OOD domain for different SVIRO domains and incompetence scores. As the threshold for the competence regions, we choose the 95% percentile of the ID validation set. For all metrics, a higher value means better performance (↑). All displayed values are medians over different domain roles and classifiers, brackets indicate 90% confidence interval.

10 epochs on DomainNet and SVIRO. When no improvement in terms of accuracy on the validation set is achieved, we stop the training. The best model is chosen due to the accuracy on the ID distribution measured via the accuracy on the validation set.

Some scores are computed on the logits and some on the features. If computed on the features, we use the output of the penultimate layer of the model as input to the score function.

We distinguish between training, validation, and test set of the ID distribution. For the OOD distribution we only consider one data set provided by the DG task which is not seen during training. Score quantiles are always computed on the ID validation set. The ID accuracy is computed on the ID test data set. If score functions need optimization (as with GMM), we train them on the ID training set. If a score function needs optimization, we restrict the training set to 50 000 samples. This only affects the DomainNet data set. We do only little to no optimization of the parameters of the score functions. We mainly stay in line with the standard settings found in the literature. For Deep-KNN we choose $K = 1$ because it shows slightly improved performance on the ID distribution (only inspected on PACS).

| ID data set | Open world data set | Test domain → Open world domains | Open world classes |
|---|---|---|---|
| PACS | DomainNet | art → painting
cartoon → clipart
photo → photo
sketch → sketch | alarm clock, ambulance, apple, backpack, baseball, basketball, bat, bear,bed and bicycle |
| VLCS | PACS | all environments → photo | elephant, giraffe and guitar |
| OfficeHome | DomainNet | art → paint
clipart → clipart
product → real
real world → real | bread, butterfly, cake, carrot, cat |
| TerraIncognita | PACS | all enviroments → photo | elephant, giraffe and horse |

Table 5: Open world extensions of different DG data sets and their test domains.

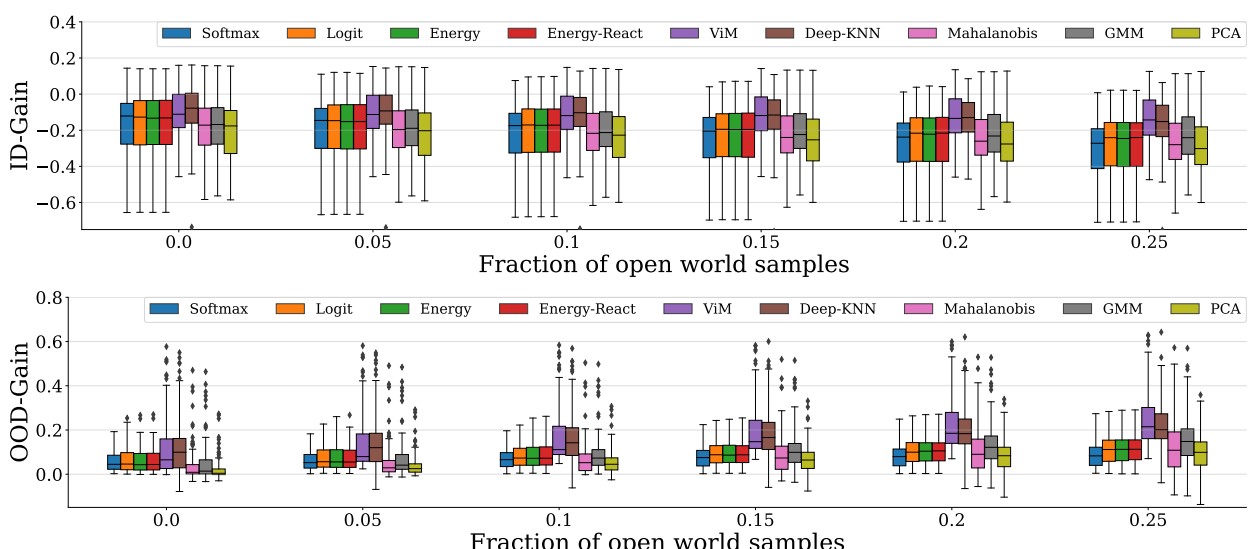

Figure 17: *OOD-Gain* and *ID-Gain* for different incompetence score for an increasing fraction of open world data (unknown classes) in the test domain (higher is better).

## A.6 Trained Classifiers

Figure 20 shows the accuracies of all different Classifiers on all DG data sets for the ID data and the OOD data. Here we show the means and standard deviations over the different domains. All classifiers obtain a similar ID and OOD accuracy. One exception is VREx which did not converge for all domains on DomainNet. In Figure 19 we show the accuracies for the different DG methods on all data sets.

## A.7 Proof of Proposition 3.1

*Proof.*    (b) Take the limit $X_C(\alpha) \xrightarrow{\alpha \to \infty} \mathbb{R}^D$. Then there is no restriction of the support of $p_{\text{OOD}}$, so the accuracy for large $\alpha$ approaches the accuracy on the entire OOD data set.

(a) By assumption $p_{\text{ID}}$ and $p_{\text{OOD}}$ share their support when restricted to the competence region $X_C(\alpha)$ when $\alpha \leq \alpha^*$. Thus we can always assume that $p_{\text{OOD}}(X_C(\alpha)) > 0$ for all $\alpha \geq \min_{x \in \text{supp}(P_{\text{ID}})} s_c(x) =: \alpha_0$, which makes the accuracy well-defined for all relevant $\alpha \geq \alpha_0$:

$$A_{\text{OOD}}(\alpha) = \frac{P_{\text{OOD}}(X_C(\alpha), c(X) = Y_{\text{true}})}{P_{\text{OOD}}(X_C(\alpha))}. \tag{3}$$

Here, $Y_{\text{true}}$ is the correct label to the input $X$.

For the remainder, we consider $\alpha \in [\alpha_0, \alpha^*]$, so $A_{\text{OOD}}(\alpha) = A_{\text{ID}}(\alpha)$. Then, we have that $A_{\text{OOD}}(\alpha) = A_{\text{ID}}(\alpha) \geq \lim_{\alpha \to \infty} A_{\text{ID}}(\alpha) = A_{\text{ID}}$. The limit can be taken analogously to the proof of (b) above.

□

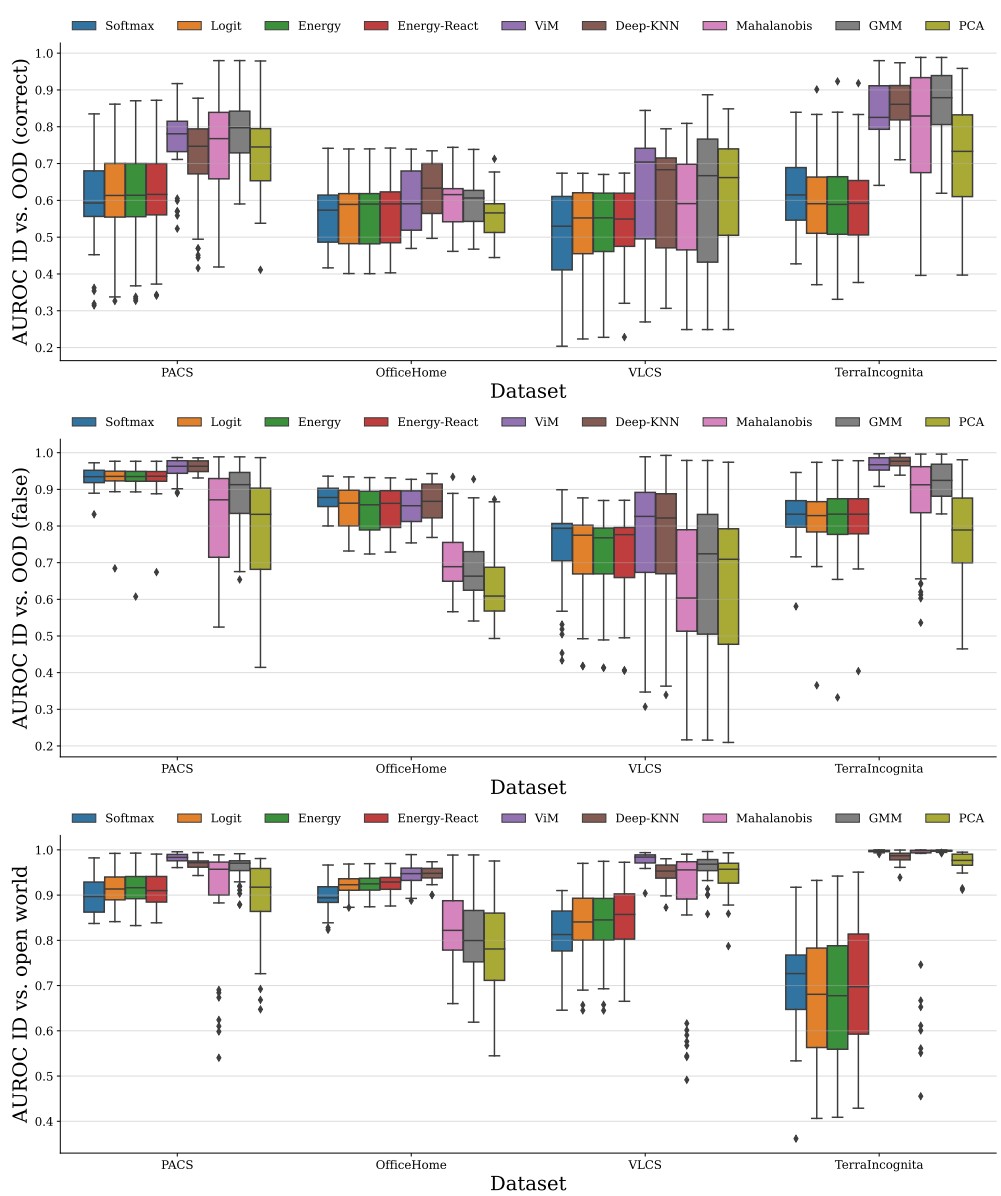

Figure 18: *Above:* AUROC of delineating ID data vs. correctly classified samples on the OOD data. *Middle:* AUROC of delineating ID data vs. wrongly classified samples on the OOD data. *Below:* AUROC of delineating ID data vs. open world data in general. All test domains are enriched with 25% open world outliers.

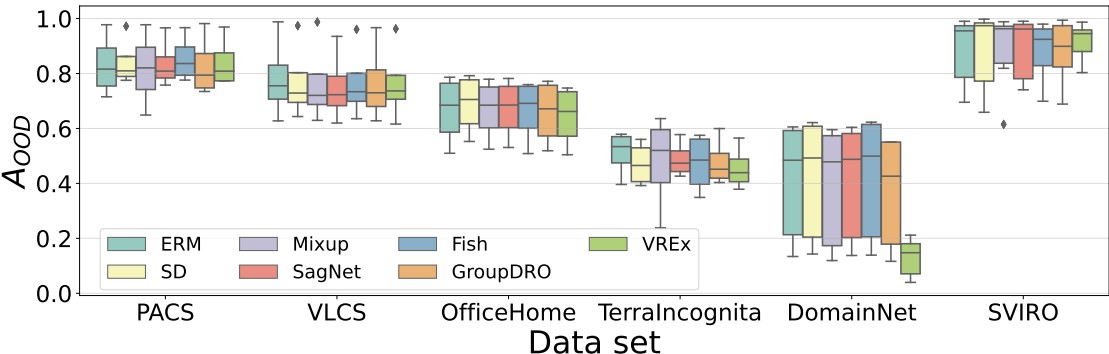

Figure 19: Accuracies for different classifiers on OOD test data. The boxes show the quartiles and medians.

Figure 20: Accuracies for different classifiers on ID and OOD test data. We show the means and standard deviations over different DG tasks.

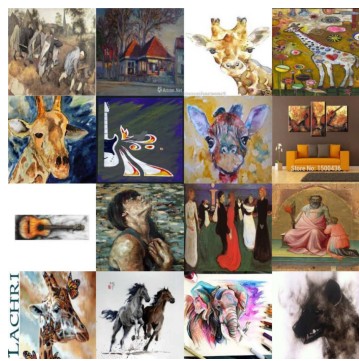

(a) Test data for domain Art

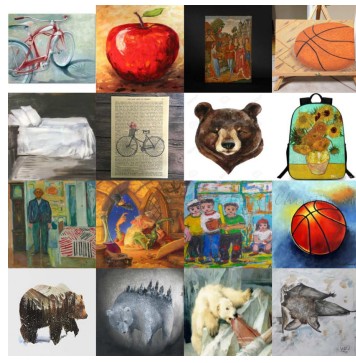

(b) Open world data for domain Art

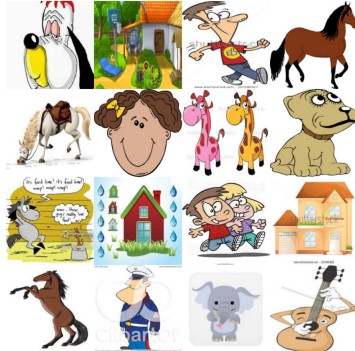

(c) Test data for domain Cartoon

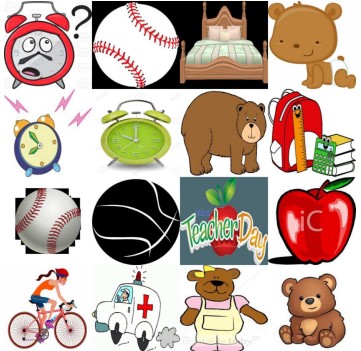

(d) Open world data for domain Cartoon.

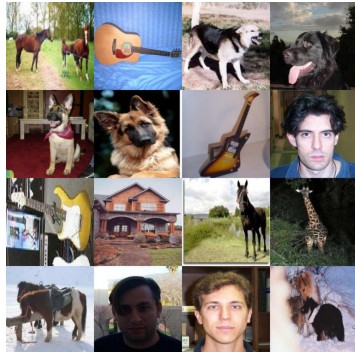

(e) Test data from domain Photo.

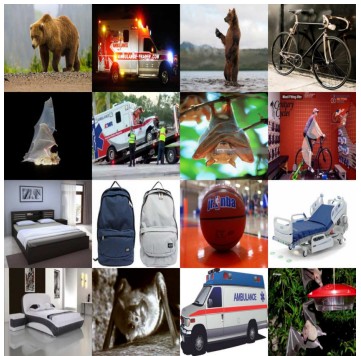

(f) Open world data for domain Photo.

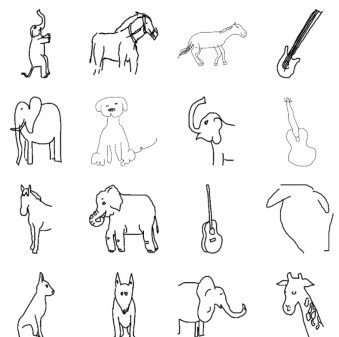

(g) Test data from domain Sketch.

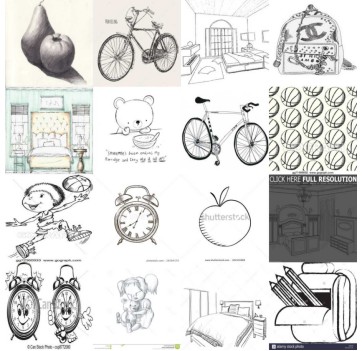

(h) Open world data for domain Sketch.

Figure 21: Test (left) and open world data (right) for the PACS data set.

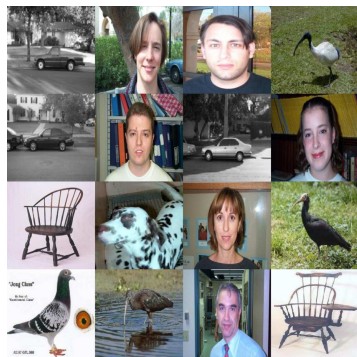

(a) Test data from domain Cal101.

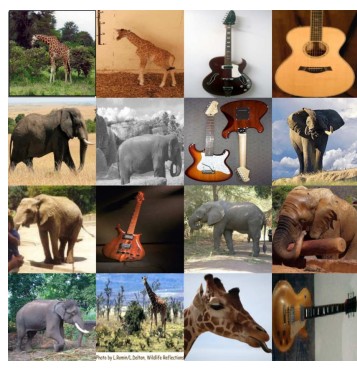

(b) Open world data for domain Cal101.

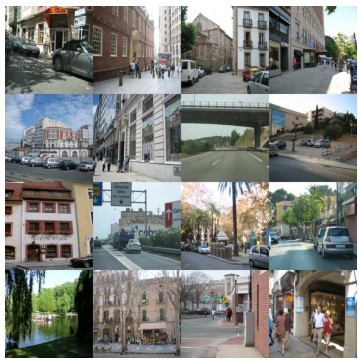

(c) Test data from domain LabelMe.

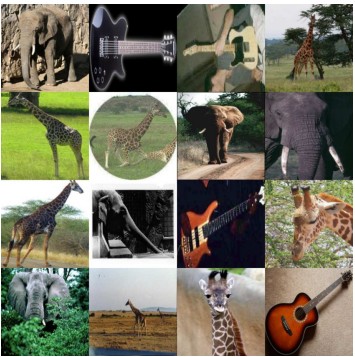

(d) Open world data for domain LabelMe.

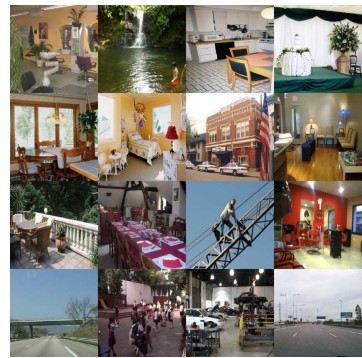

(e) Test data from domain SUN09.

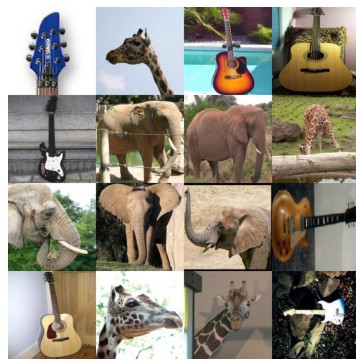

(f) Open world data for domain SUN09.

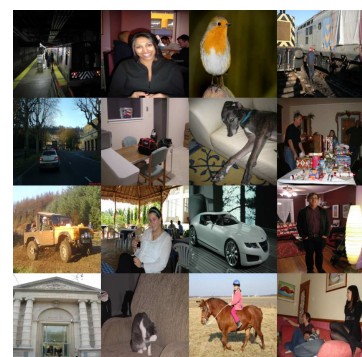

(g) Test data from domain VOC2007.

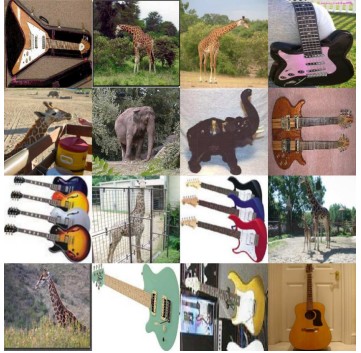

(h) Open world data for domain VOC2007.

Figure 22: Test (left) and open world data (right) for the VLCS data set.

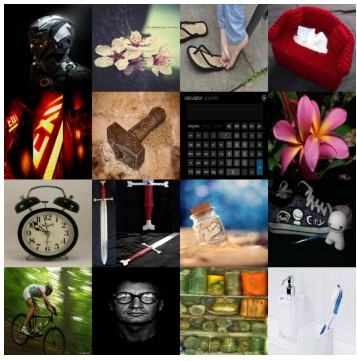

(a) Test data from domain Art.

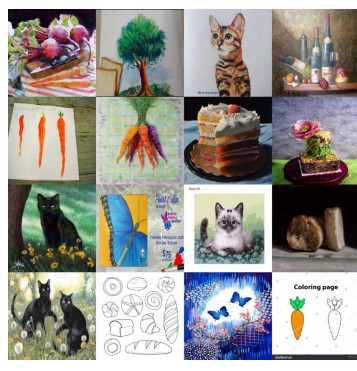

(b) Open world data for domain Art.

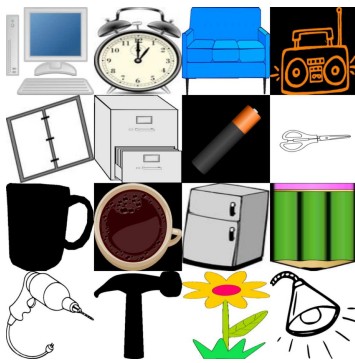

(c) Test data from domain Clipart.

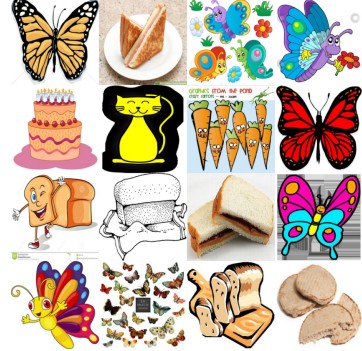

(d) Open world data for domain Clipart.

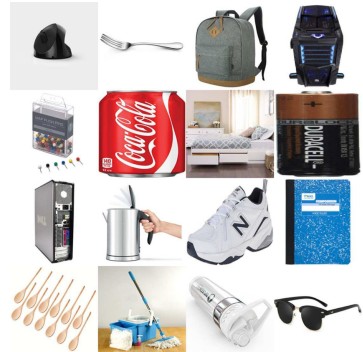

(e) Test data from domain Product.

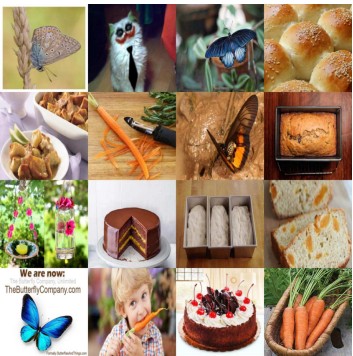

(f) Open world data for domain Product.

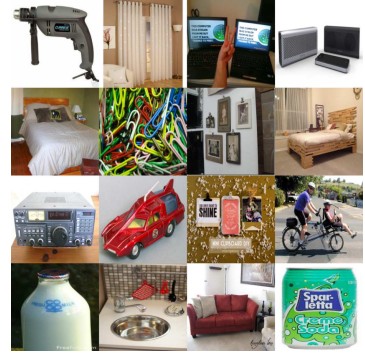

(g) Test data from domain Real World.

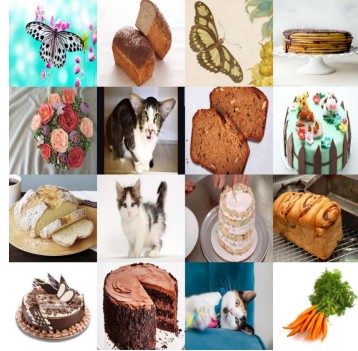

(h) Open world data for domain Real World.

Figure 23: Test (left) and open world data (right) for the OfficeHome data set.

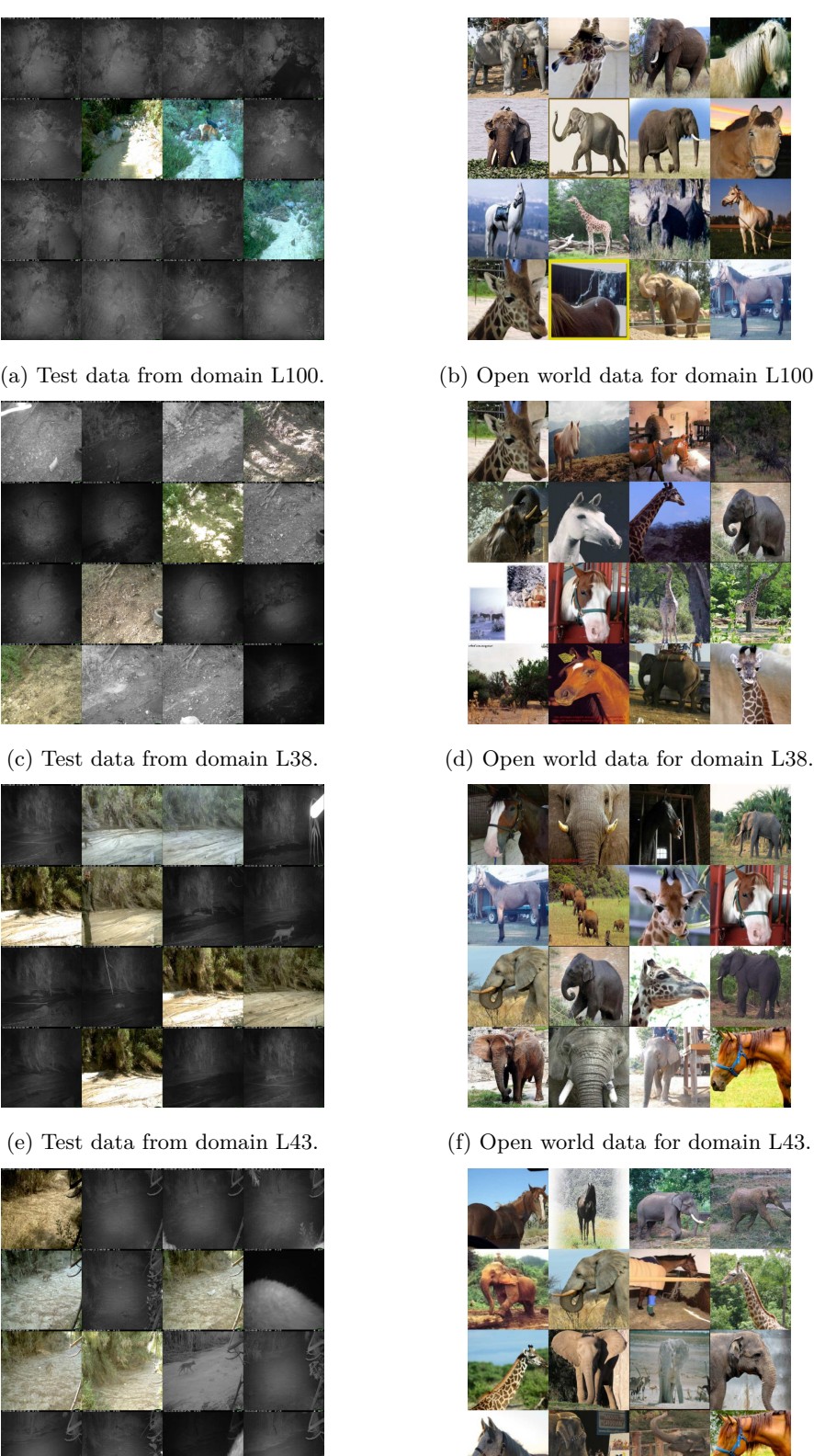

(a) Test data from domain L100.

(b) Open world data for domain L100.

(c) Test data from domain L38.

(d) Open world data for domain L38.

(e) Test data from domain L43.

(f) Open world data for domain L43.

(g) Test data from domain L46.

(h) Open world data for domain L46.

Figure 24: Test (left) and open world data (right) for the TerraIncognita data set.

