# OpenReview forum: "Finding Competence Regions in Domain Generalization"
_TMLR — Accepted by TMLR_

### Review · Reviewer_hwUH · 2023-04-09

**Summary Of Contributions:**

The paper performs an empirical analysis of the performance of various existing methods for "OOD detection". This is in the context of using these methods to abstain on some fraction of the test points in the hope that average accuracy on the non-abstained points will increase.

**Audience:**

No

**Claims And Evidence:**

No

**Requested Changes:**

- The paper should be clearer about what it is *contributing as new* versus what it is *building upon*. The work does not present a new framework, nor does it suggest a new method to "compare with prior work". It is an empirical evaluation of existing approaches. This is a valuable contribution! But it must be honestly presented as such.

- The references and related work need quite a bit of work to give credit appropriately. Simply citing one recent survey or famous paper from the past 3 years is not sufficient.

- I think the work would be greatly improved by dropping Proposition 3.1, and presenting the definition as a desideratum (which is unachievable, but may be something to strive for). In general, the math here doesn't really add to the paper.

- The idea of thresholding a "competence score" such as confidence or some other measure of "OOD-ness" already exists. Rather than spending so much time rehashing it (see the first point above), the paper would benefit from more time spent presenting experimental results, which are the meat of the contribution.

- While the number of experiments run is large, the actual value to be gained from these results feels quite minimal. A more thorough evaluation which explores something more applicable to real-world use cases feels apropos and would strengthen the results.

**Strengths And Weaknesses:**

I have several major concerns with this paper, most of which I believe are fixable, but probably not in this submission cycle. I encourage the authors to take the time to redesign the paper, reframe its contributions, and possibly resubmit, but I cannot recommend for acceptance currently, nor would I feel comfortable doing without major revisions.

1. The earlier sections (even the abstract) belie the actual contribution of the work. The paper is framed as "introducing" this idea of abstention ("learning to reject") despite there being a long history of existing work on this (see point 2 for discussion on references). Simply giving a new name ("competence score") to an existing concept is not grounds for claiming to introduce a new framework, which is exactly what the first line of the abstract does. What this paper *does* contribute is an empirical analysis of several existing methods. The use of phrases with "For comparability with prior work" are misleading and suggest that this paper is presenting a new idea or method.

The line "We present a comprehensive experimental evaluation of incompetence scores for classification and highlight the resulting trade-offs between rejection rate and accuracy gain" is reasonably accurate, in that it begins to describe what it is that this paper actually does. But the first five pages of the paper discuss things that have already been done, and most of the experiments are relegated to the Appendix. Since this paper is an experimental study, why not spend more of the paper presenting and discussing actual findings?

2.  References to prior work are *exceptionally* sparse and poorly researched. Several times the paper references long lines of work with a great deal of literature from as far back as half a century ago (e.g. Chow, 1970) by simply citing a single survey from the past few years. The Related Work section is a little better, but still leaves a lot to be desired. As an extreme example, I note that the first subsection which discusses OOD Detection and Generalization **does not cite a single source from before 2021**. Also, why is Gulrajani and Lopez-Paz (2020) given as the source for Domain Generalization in the intro?? This work did not introduce or popularize the problem, nor did it provide a meaningful survey. The paper is full of such questionable citations. The section on Selective Classification is better, but the only citations from before 2020 are a line of work by a single author, further indicating the lack of depth to this discussion.

3. Definition 3.1 is unachievable. What is the value in a formal definition of this sort if the thing being defined does not exist? It also uses the phrase "for any distribution of interest." What does this mean? Is this a formal definition or isn't it? Because that expression doesn't actually mean anything precise. The small amount of math in this paper strikes me as the classic example of math for math's sake---Proposition 3.1 doesn't seem to be saying anything that isn't just as clear in plain English. I think this paper would be much better served fully committing to the "empirical evaluation" angle.

4. Finally, I question the meaningfulness of the experimental evaluation: I think such a direction *could* be quite interesting, but it doesn't feel that way in its current form. In particular, the fundamental problem of OOD generalization is that for any given task, one *does not know* the best competency score to use, nor the threshold to set. Rather than fixing $\alpha=.95$, why not explore a range of values?

While the number of experiments run is large, to me the actual value to be gained from these results feels quite minimal. I note that doing a "post-hoc" analysis is not very helpful, unless it shows that a particular method is universally superior, which is basically never going to be the case. I encourage the authors to consider the following alternative analysis which I think would be much more informative: given a set of training environments, do LOOCV to **select** the best competence score and/or threshold according to one's desired maximum rate of abstention. **Then**, use this to evaluate on the held-out domain. This study would give a meaningful demonstration of what to expect on a *future, real-world* distribution shift where we do not have access to the test domain and therefore cannot identify what is the correct competence score or threshold to be using. An important question to consider when framing the current results: what good does it do to tell us that the quality of the $\alpha=0.95$ threshold depends on the competence score and the distribution? Is this not already obvious, given the difficulty and variety of distribution shift in general? Why is it helpful to show that for one specific benchmark, a specific pattern emerges, if this pattern won't be true for future shifts?

As I said before, all of these issues can be remedied---but they are large enough in the current submission that I can't recommend acceptance, even keeping in mind the TMLR guidelines.

---

> ### Author Response · Authors · 2023-05-04
> **Response to Reviewer hwUH (1/2)**
>
> We thank the reviewer for the valuable and helpful feedback. We are pleased to hear that the reviewer recognizes the significance of our contribution and we hope that we can address the concerns with our revised version.
>
> **Contribution and Embedding in Prior Work**
>
> The reviewer has correctly identified that we do not introduce a new learning to reject framework but rather investigate the merits of such a framework for addressing silent failures in Domain Generalization (DG) contexts. We acknowledge that this goal was not clearly delineated in the original version of the manuscript. Thus, we have updated the Introduction and Abstract accordingly, emphasizing our particular contributions in the appropriate DG setting.
>
> To summarize: In the Abstract, we have changed the verb ‘propose’ to ‘investigate’ to avoid confusion. We have also emphasized that we use existing proxy scores. The second paragraph in the Introduction now highlights the intersection between Domain Generalization and learning to reject, specifically noting that out-of-distribution (OOD) samples are commonplace rather than exceptional. In the third and fourth paragraphs of the Introduction, we now discuss the unique aspects of this intersection, such as the challenge to determine an appropriate threshold. In the fifth paragraph, we connect the competence region to prior work and explain our approach. We believe that the new changes underline the distinction between the novel aspects of our paper and their relationship with prior research.
>
> **More Discussion for Experiments**
>
> The reviewer’s remark is reasonable. Accordingly, we extended our experimental findings in the updated version of the manuscript. In Section 4.3, we conducted an investigation into the trade-off between accuracy and coverage. Additionally, we included a box plot that illustrates that no domain-robust classifier performs consistently better than the naive baseline (Empirical-Risk-Minimization).
>
> However, we still think that it is necessary to introduce the formal framework, wherein the experiments should be interpreted, especially in view of the abundant and terminologically saturated prior work. Indeed, as Reviewer zTpB pointed out, a clearer method section is needed to make this work more accessible to many readers. Nevertheless, we condensed the methods section to its essential components and made an effort to avoid unnecessary elaboration on already established information (e.g., the trade-off between accuracy and coverage).
>
> **References to prior Work**
>
> We have thoroughly re-worked and augmented the Related Work section with a more comprehensive context, which not only covers crucial prior research in Domain Generalization, selective classification, and OOD detection, but also earlier significant research (see Section 2).
>
> The Gulrajani and Lopez (2020) citation in the Introduction should refer to the DomainBed datasets. Regarding DG we now reference the paper that introduced the term DG (Muandet et al., 2013) and a more recent survey paper on DG (Zhou et al., 2022).
>
> We are happy to incorporate additional sources if the reviewer has particular ones in mind.
>
> **Definition and Proposition**
>
> We believe that it is crucial to establish the conditions in which we anticipate the incompetence score to be beneficial. Furthermore, in practice, we have observed that the "Definition" is fulfilled (cfg. Figure 3). However, we acknowledge the potential for confusion, therefore we have decided to refer to it as a Criterion rather than a Definition. In regards to Proposition 3.1, we agree that while it may not be overly complex, it does rely on certain non-obvious conditions, such as having the same support space. The proposition provides insight into the behavior of the accuracy curve computed on OOD data, and we believe that it is a non-trivial statement. In line with your suggestion, we condensed Section 3.2 considerably.
>
> **Threshold**
>
> We thank the reviewer for the suggestion to explore a range of threshold values. We have added a new Figure 4 that demonstrates the accuracy and coverage trade-off for the classifiers in various domains across all percentiles, while also displaying the fixed 0.95 percentile.
> However, it is worth mentioning that the coverage-accuracy curves, as depicted in Figure 4, are not available when novel domains are encountered.
>
> Given that new data may also emerge within the same domain as the training data, the 95th percentile can be interpreted as ensuring a level of coverage for such In-Distribution (ID) data.
> Aiming for a high level of specificity at a preset sensitivity level of around 0.95 is standard (although admittedly conventional) approach in the statistical literature, and it aligns with our threshold. We recognize that there is a degree of subjectivity involved, so we also explored thresholds close to the 95th percentile and demonstrated that the relative performance of the scores remains quite consistent (see Appendix A.2; Figure 16)

---

> > ### Author Response · Authors · 2023-05-04
> > **Response to Reviewer hwUH (2/2)**
> >
> > **Gained Insights**
> >
> > We would like to draw the reviewer’s attention to Section 4.6 (previously 4.5), where we attempted to learn a threshold on the ID data set. While this approach was more effective in achieving the ID-Accuracy on the novel domain compared to setting the threshold to the 0.95 percentile, it still encountered failure cases, indicating the fact that the utility of OOD scores may not transfer seamlessly to new domains. We agree that your proposed analysis might yield additional insight, but we have to leave it to future work due to the high computational requirements (a multiple of the current computational budget).
> >
> >
> > Further, Reviewer CkVU highlights the value of our experiments as a strength, namely that  “The experiments provide some useful insights. In particular: the demonstration of the trade-off between coverage and accuracy, the utility of different incompetence scores, and analysis in the open world setting.” We also think that we can show interesting aspects and can draw useful insights from the experiments. We would also like to mention that the

---

### Review · Reviewer_CkvU · 2023-04-17

**Summary Of Contributions:**

The paper proposes a novel problem setting that combines aspects of domain generalisation with the prediction with option to reject setting. The submission investigates how well existing scores for detecting out of distribution examples can be adapted to estimate the incompetence of a model on data that is guaranteed to be out of distribution, but possibly similar enough to still be correctly labelled by the model. It is found that accuracy of predictions where the classifier was estimated to be competent are much higher than those predictions where the classifier is presumed to be incompetent. As expected, there is a trade-off between accuracy and coverage. Other experiments investigate an open world variant of the new problem setting, and a heuristic for setting the threshold to be applied to the incompetence score.

**Audience:**

Yes

**Broader Impact Concerns:**

No significant broader impact concerns.

**Claims And Evidence:**

Yes

**Requested Changes:**

* The inroduction of the new problem setting could be made more explicit. The paper does not actually spell out the main differences between their proposed problem setting and the two most closely related previously studied settings: DG and prediction with option to reject.
* There is too much going on in Figure 3. It Would be better to split this up into a few different figures, rather than trying to make several points with the same figure.
* A more direct illustration of the trade-off between coverage and accuracy would be good. For example, plots with coverage on the x-axis and accuracy on the y axis would illustrate this. One could construct these in a similar way to precision--recall curves or ROC curves, by evaluating coverage/accuracy for a large number of different incompetence score thresholds. The area under this curve would also provide a single metric for doing evaluation in a threshold-free manner.

**Strengths And Weaknesses:**

* The new problem setting is interesting and I think it is a sensible combination of existing settings.
* The notion of an $\alpha$ competence region seems quite useful, and the idea of constructing them via existing OOD detectors and some percentile of the in-domain data seems sound.
* The experiments provide some useful insights. In particular: the demonstration of the trade-off between coverage and accuracy, the utility of different incompetence scores, and analysis in the open world setting.
* Choose 5% as the acceptance threshold seems a bit arbitrary. It would have been interesting if the paper had investigated the sensitivity of this hyperparameter a bit more. I could see different tasks needing different thresholds here, and it would be nice if this was discussed a bit more.
* The main weakness of the paper in my view is that the presentation could be significantly improved. I have provided several concrete suggestions below, but I suggest the authors make a pass over the paper and try to spell out some of the implications of their results a bit more clearly.

---

> ### Author Response · Authors · 2023-05-04
> **Response to Reviewer CkvU**
>
> We express our gratitude to reviewer CkvU for the thorough review and valuable feedback provided.
>
> **Abstract and Introduction**
>
> Thank you for pointing out the need for a more explicit delineation of our problem setting. We have updated to Introduction and Abstract to emphasize the concrete features of our problem setting and augmented the Related Work considerably.
>
> To summarize: The second paragraph in the Introduction now highlights the intersection between Domain Generalization and learning to reject, specifically noting that out-of-distribution (OOD) samples are commonplace rather than exceptional. In the third and fourth paragraphs of the Introduction, we now discuss the unique aspects of this intersection, such as the challenge to determine an appropriate threshold. In the fifth paragraph, we connect the competence region to prior work and explain our approach. We believe that the new changes underline the distinction between the novel aspects of our paper and their relationship with prior research.
>
> **Figure 3**
>
> We have reduced Figure 3 to its essential parts which should now be more understandable.
>
> **Trade-Off Coverage/Accuracy**
>
> The reviewer proposed a very illustrative and interesting idea to visualize the coverage-accuracy trade-off. We have added such a graph in Figure 5, with the inclusion of the 95%-Percentile threshold to illustrate their positions on the accuracy-coverage spectrum.

---

### Review · Reviewer_zTpB · 2023-04-23

**Summary Of Contributions:**

The task addressed in the article is Domain Generalization, where the distribution of the test data is different from that of the training data. The problem is to ensure that the trained model can perform well on the test data despite the domain shift, without requiring retraining on new data.

The authors attempt to address the task of domain generalization where the test data’s distribution differs from that of training. The authors propose a method that learns to reject samples based on an “incompetence threshold” empirically showing improved accuracy on the task of classification. The threshold is determined by the in-distribution data (seemingly coming from the training set) which is used to reject samples based on either logits-based incompetence score or feature-based incompetence score. The authors also claim to show experiments where robust classifiers failed in comparison to basic baselines.

**Audience:**

Yes

**Broader Impact Concerns:**

I do not see Ethical Implications of this work.

**Claims And Evidence:**

No

**Requested Changes:**

I can't recommend an acceptance unless changes are made to address the Weaknesses mentioned in the previous text box.

**Strengths And Weaknesses:**

Strengths:
- The motivation of this task is well written and the authors created an enjoyable story line throughout the paper, especially the Introduction.
- The authors did a great job identifying different works to compute scores based on Feature-based, Density-based, Reconstruction-based,  and Logits-based methods
- The authors compare these methods across different datasets and showcase their behavior which might be useful for practitioners who would like to use similar methods to maximize performance on similar datasets


Weaknesses:
The Method Section needs more details to be understood. I was confused about how the "Incompetence Score" was computed until I started reading the Experiment Sections. I would encourage the authors to either reference how the scores are computed by adding a citation to the Experiment Section, or putting how the scores are computed in the Method section explicitly.

Figure 3 is confusing, what is the x-axis value? it seems like it is alpha, but isn’t that the percentage of accepted samples?

It is very difficult to identify which Figure and Table support the statements in the Contributions listed at the end of the Introduction. For example, I still don’t know which Figure/Table shows that robust classifiers are worse than the naive baseline. I strongly suggest for the authors to add a citation of the Table and Figure in front of each of the 4 contributions listed. Otherwise I can't verify them.

The authors mentioned that their method also works with Regression but they do not explain how.

The contribution that there is a tradeoff between alpha and the classification performance is obvious. That  tradeoff is commonly observed with self-training methods in the semi-supervised learning literature [A] where picking samples using a certain confidence threshold such as   95% and above could lead to improved accuracy, but lowering that confidence threshold results in picking up bad samples which hurt performance. So this contribution that the authors list is very weak.

The authors should definitely include many of the semi-supervised literature where samples are selected based thresholds such as the ones here [A]  and the works that cite it.  The authors should also compare so some of these self-training methods especially the ones that pick examples using a confidence threshold.

The novelty in this work is very limited as the authors do not propose a new method but simply compare existing methods on few datasets.

The analysis made by the authors is very limited and difficult to draw conclusions as the datasets are small. Therefore the conclusions that Feature-based score is better than logits-based score is better for OOD cannot be verified unless more experiments are conducted on larger datasets like ImageNet.

Specifying the percentage as 95% is very specific and I doubt that it works well on different problem setups. Further, blindly doing a cut-off at 95% could lead to an imbalanced dataset in the case where one class is easy to classify and the other classes are more difficult leading to  selecting samples only from the easy classes leading to a biased dataset.

[A] Zhu, X. 2006. Semi-Supervised Learning Literature Survey.

---

> ### Author Response · Authors · 2023-05-04
> **Response to Reviewer zTpB**
>
> We thank Reviewer zTpB for taking the time to review our paper and giving helpful feedback.
>
> **Suggestion: Score introduction in Method section.**
>
> This is a good idea. We moved the scores from the experimental Section 4.1 to the method Section 3.1. in order to clarify the computation of the scores early on.
>
> **Missing references of Contributions in Introduction**
>
> We have included a reference for each of our contributions, including a figure that illustrates that a naive baseline is not inferior to the DG methods (see Section 4.4 and Figure 5).
>
> **Figure 3**
>
> Thank you for this remark. We have added the x-axis value and condensed the figure to highlight its essential elements.
>
> **Regression Setting**
>
> We do not expand on the topic of regression in order to follow a single and consistent narrative in the current paper that already includes quite a few moving parts (e.g., different classifier architectures, data sets, domains, etc.). Moreover, not all scores investigated in this work can be transferred to the regression setting. However, we now remark in the Introduction that “incompetence scores computed on the feature space as proxy scores for the regression error are similarly applicable in this case”, and leave the systematic investigation of the regression setting to future research.
>
> **Self-Training and Semi-Supervised Learning**
>
> While the concept of employing self-training is intriguing, it falls outside the scope of Domain Generalization. Within the DG framework, it is assumed that there is no access to test data during training. Using unlabeled data for training would necessitate having unlabeled data from the test domain, which would classify it as belonging to the Domain Adaptation category. We made sure to provide a clearer distinction between Domain Adaptation and Domain Generalization in the related work section.
>
> **Novelty**
>
> As pointed out by the other reviewers, one of our main contributions is to identify a sensible scenario (i.e., Domain Generalization) where selective classification can be beneficially applied to, and to systematically investigate this scenario. We updated the Abstract and Introduction for more clarity on this point.
>
> **Dataset Size and Conclusions**
>
> We believe that our datasets are not small; for instance, DomainNet comprises roughly 600k images across all domains. In total, we conduct numerous experiments, covering 32 distinct training scenarios via various leave-one-domain-out procedures. It is worth mentioning that Reviewer hwUH observes that “the number of experiments run is large”, while Reviewer CkvU acknowledges that the “experiments provide some useful insights”. Furthermore, we would like to emphasize that ImageNet is not a Domain Generalization dataset, as it does not offer distinct domains. Hence, it falls outside the scope of the evaluation in our study.
>
> **Threshold**
>
> Given that new data may also emerge within the same domain as the training data, the 95th percentile can be interpreted as ensuring a level of coverage for such In-Distribution (ID) data.
>
> Aiming for a high level of specificity at a preset sensitivity level around 0.95 is a standard (although admittedly conventional) approach in the statistical literature, and it aligns with our threshold. We recognize that there is a degree of subjectivity involved, so we also explored thresholds close to the 95th percentile and demonstrated that the relative performance of the scores remains quite consistent (see Appendix A.2; Figure 16)
>
> From our experiments, it became apparent that configuring the 95th percentile resulted in issues, such as a significant decline in accuracy within the competence region. We would like to draw the Reviewer’s attention to Section 4.6 in which we also propose to learn a threshold which might work better for a broader set of situations. However, due to the domain shift, we have also observed instances of failure and cannot provide guarantees for this tentative approach.
>
> **Class Imbalance**
>
> From our perspective, acquiring an imbalanced dataset during testing is not a drawback, but rather an advantage. Given that the labels are unknown during testing, selecting only the classes that are straightforward to classify can potentially enhance performance and reliability. This approach could also be advantageous in a medical context, where, for instance, a predictive model accurately predicts the easy classes and leaves the difficult ones to be addressed by a medical professional.
>
> Further, note also, that (highly) imbalanced data sets pose a primary problem during training. Thus, the reviewer’s point is valid if one is to re-train the model on an imbalanced data set resulting from our rejection method, that is, in a Domain Adaptation setting.

---

### Author Response · Authors · 2023-05-04
**Revised Version**

We thank all Reviewers for their valuable and helpful feedback. All reviewers agree that the topic of the paper is interesting and worth pursuing. In our individual responses, we have covered all reviewer comments and made significant changes to the text accordingly. Updated parts in the document are highlighted in blue. The major changes are:

* We have thoroughly revised the Introduction to emphasize the particular combination of issues addressed in this study, thereby highlighting the distinction between the novel aspects and their relationship with prior research.

* We have augmented the Related Work with a more comprehensive context, which not only covers crucial prior research in DG, selective classification and OOD detection, but also much earlier research (see Section  2).

* We have examined the trade-off between coverage and accuracy in greater detail in Section 4.3

* We have distilled Figure 3 into its essential parts with the hope to enhance its readability.

* We have moved the description of the scores from the experimental Section 4.1 to the methods Section 3.1.

Finally, we would like to point out that one of our central results has important practical implications, namely, that none of the scores is able to achieve a satisfactory competence region across all domain generalization tasks. Thus, safety-critical applications should not fully rely on a single proxy score for assessing out-of-domain competence.

---

### Decision · Action_Editors · 2023-05-30

**Recommendation:** Accept as is

**Comment:**

Please see the summary of the strength and (addressed) weakness as above. Don't forget to consider reviewer hwUH's minor comments in the camera-ready version.

**Audience:**

Yes

**Claims And Evidence:**

The paper addresses the task of domain generalization, where the distribution of test data differs from that of training data. The proposed method learns to reject samples based on an "incompetence threshold" to improve classification accuracy. The strengths of the paper lie in the well-written motivation, comprehensive analysis of different incompetence scores, and empirical insights into trade-offs between rejection rate and accuracy gain. The paper provides useful findings for practitioners working on similar datasets.

In the first version, the presentation needed significant improvement, as the framing of the contribution as "introducing" a new idea is misleading, given the existing work on abstention. The paper lacked in-depth references to prior work, and citations are often questionable. The formal definitions and mathematical propositions did not seem to add much value and could be replaced with a stronger focus on empirical evaluation. The experimental evaluation, although extensive, did not provide significant value due to the lack of exploration of a range of values and a meaningful analysis.

In the discussion, all the reviewers are satisfied with the revision and the author's response.

---

> ### Author Response · Authors · 2023-05-31
>
> We express our gratitude to the Action Editor for approving our paper for publication in TMLR, and we extend our appreciation to all the reviewers for their valuable comments, which lead to substantial improvements in our revised version. We would also like to incorporate the final (supposedly minor) comments by reviewer hwUH in our camera-ready version, but are currently unable to see them. Could you unlock those comments?

---

> > ### Comment · Reviewer_hwUH · 2023-06-02
> > **here they are**
> >
> > Sorry, I did not realize that my recommendation was not visible to the authors. I've copy-pasted below
> >
> > "In my original review, I wrote that I would hope to see major revisions before recommending acceptance. I'm happy to see that the authors have made a large number of changes addressing most of my concerns, and with the new experiments as well I think this meets the bar for acceptance.
> >
> > One point, while Definition 3.1 was changed to a criterion, the issue I was trying to raise was how vague it is, in particular the expression "for any distribution of interest". I don't think this could possibly ever be achieved unless the incompetence score is defined as "accuracy on the distribution of interest". Figure 3 is promising, but the whole difficulty of distribution shift is in black swan events which don't match existing benchmarks. So I think using this kind of terminology is a bit sketchy, and I'd still recommend rephrasing."